# Context-dependent function of TSLP and IL-1β in skin allergic sensitization and atopic march

Justine Segaud [1,5], Wenjin Yao [1,5], Pierre Marschall [1], François Daubeuf [2,3], Christine Lehalle[2,3], Beatriz German[1], Pierre Meyer [1], Pierre Hener[1], Cécile Hugel [1], Eric Flatter[1], Marine Guivarch [1], Laetitia Clauss[1], Stefan F. Martin [4], Mustapha Oulad-Abdelghani[1] & Mei Li [1] ✉

Atopic diseases, including atopic dermatitis (AD) and asthma, affect a large proportion of the population, with increasing prevalence worldwide. AD often precedes the development of asthma, known as the atopic march. Allergen sensitization developed through the barrier-defective skin of AD has been recognized to be a critical step leading to asthma, in which thymic stromal lymphopoietin (TSLP) was previously shown to be critical. In this study, using a laser-assistant microporation system to disrupt targeted skin layers for generating micropores at a precise anatomic depth of mouse skin, we model allergen exposure superficially or deeply in the skin, leading to epicutaneous sensitization or dermacutaneous sensitization that is associated with a different cytokine microenvironment. Our work shows a differential requirement for TSLP in these two contexts, and identifies an important function for IL-1β, which is independent of TSLP, in promoting allergen sensitization and subsequent allergic asthma.

Atopic diseases, including atopic dermatitis (AD) and asthma, affect a large proportion of the population, with increasing prevalence worldwide. These diseases result in considerable morbidity, and are in some circumstances life-threatening, raising these diseases as major health problems. AD is a chronic inflammatory skin disease characterized by defective skin barrier, cutaneous inflammation with infiltration of Th2 cells, eosinophils as well as expression of Th2 cytokines and allergen-specific immunoglobulin (Ig) E production[1,2]. It usually starts in early infancy and precedes other atopic diseases such as asthma. More than 50% of moderate to severe AD children develop asthma and/or allergic rhinitis at a later stage, called the atopic march[3–6]. In addition to the efforts in the development of medical therapy for asthma, it is critically important to develop strategies to prevent and block the atopic march.

It has been found that AD children are prone to develop allergen sensitization, indicated by T cell and B cell memories to allergens. Cutaneous exposure to allergens, such as aeroallergen house dust mites (HDM) is recognised as a critical route for sensitization in AD patients[7]. It is thus assumed, which has been supported by studies from mouse models[8,9], that following the phase of skin sensitization, allergen challenge upon re-encountering of the allergen in the airway at a later stage results in the development of asthmatic symptoms. Recently, a human cohort study showed that AD with allergen sensitization has a higher risk of asthma, whereas AD without concomitant allergic sensitization is not associated with an increased risk of

[1]Institut de Génétique et de Biologie Moléculaire et Cellulaire, CNRS UMR 7104 - Inserm U 1258 – Université de Strasbourg, Illkirch, France. [2]CNRS-Strasbourg University, UAR3286, Plate-Forme de Chimie Biologique Intégrative de Strasbourg/Strasbourg Drug Discovery and Development Institute, ESBS, Illkirch, France. [3]CNRS-Strasbourg University, UMR7200, Laboratoire d'Innovation Thérapeutique/ Strasbourg Drug Discovery and Development Institute, Faculté de Pharmacie, Illkirch, France. [4]Allergy Research Group, Department of Dermatology, Medical Center – University of Freiburg, Faculty of Medicine, University of Freiburg, Freiburg, Germany. [5]These authors contributed equally: Justine Segaud, Wenjin Yao. ✉e-mail: mei@igbmc.fr

asthma[10], providing further evidence for the key role of allergic sensitization occurring during AD in the process of the atopic march. The understanding of how allergic sensitization occurs and how it is regulated in AD context is therefore crucially required for developing strategies to prevent and stop the atopic march.

It is known that AD skin not only bears a defective skin barrier allowing allergen penetration but also provides an inflammatory cytokine microenvironment conducive to the development of the sensitization to allergens[2]. We and others have previously reported that thymic stromal lymphopoietin (TSLP), a cytokine produced by skin keratinocytes, is induced by skin barrier disruption in mouse[8] or human[11], and promotes ovalbumin (OVA)-induced Th2-type sensitization through the tape-stripped skin and subsequent asthma in mice[8]. More recently, we also provided evidence that skin TSLP promotes epicutaneous OVA-induced follicular helper T (Tfh) cells, which provide critical B cell help in the germinal center (GC) of lymphoid organs[12] for the generation of allergen-specific IgE[13].

Despite of these pieces of evidence suggesting TSLP as an important target for AD therapy and for preventing the atopic march, it had not yet been explored the role of TSLP in allergen sensitization occurring in AD skin with different severity. Indeed, AD is well recognized for its heterogeneity, bearing varied skin barrier defects[14] due to various genetic or/and environmental causes, or at different stages of the diseases. Consequently, allergen exposure could occur at lesioned skin at the different anatomic depth, and be associated with different cytokine microenvironment. We thus aimed to investigate the role of TSLP in the allergen cutaneous sensitization occurring at different anatomic depth of mouse skin and the subsequent development of allergic asthma.

In this work, using a laser-assistant microporation (LMP) system to disrupt the targeted skin layers for generating micropores at a precise anatomic depth of mouse skin, we model allergen exposure superficially or deeply in the skin, leading to an epicutaneous sensitization or a dermacutaneous sensitization that is associated with different cytokine microenvironments. Our study implicates a differential requirement for TSLP in these two contexts, and identifies an important function for IL-1β, which is independent of TSLP, in promoting allergen sensitization and subsequent allergic asthma.

## Results

### TSLP is differentially required for epicutaneous or dermacutaneous HDM-induced Th2/Tfh responses

To mimic the cutaneous allergen sensitization occurring at different anatomic depth of mouse skin, we used the Precise Laser Epidermal System (P.L.E.A.S.E., Pantec Biosolutions) to fractionally ablate the targeted skin layers and generate patterned micropores, which allows us to deliver allergens to micropores at a precise depth of the skin, as recently reported[13]. As shown in Fig. 1a, the laser microporation (LMP) at the depth of 30 μm (LMP_30 μm) ablates the stratum corneum and the suprabasal layer of the epidermis of mouse ears, while the LMP at the depth of 91 μm (LMP_91 μm) ablates the epidermis and reaches the dermis. ELISA analyses showed that TSLP production was similarly induced in LMP_91μm and LMP_30 μm ears (Fig. 1b), and RNA in situ hybridization identified that in both cases TSLP expression was restricted to the epidermis, with no signal detected in the dermis (Fig. 1c), in agreement with previous reports showing that keratinocyte-derived TSLP is induced by barrier disruption in mouse[8,13] or human skin[11]. We next set out to investigate allergen sensitization occurring at different anatomic depth of the skin and subsequently the development of asthma, by establishing an experimental protocol in which house dust mite (HDM) is applied on LMP_30 μm skin (to achieve epicutaneous e.c. sensitization) or on LMP_91 μm (to achieve dermacutaneous d.c. sensitization), followed by intranasal (i.n.) challenge with HDM to induce allergic asthma (Fig. 1d).

We first analysed the Th2-type skin inflammation induced by e.c. or d.c. HDM sensitization, and compared the requirement for TSLP in these two contexts. As shown in Fig. 1e, the e.c. HDM treatment induced an inflammatory cell infiltration in the dermis of Balb/c wild-type (WT) mice, including eosinophils and basophils, two characteristic cells in Th2-type inflammation in allergic AD, which was totally abolished in $Tslp^{-/-}$ mice (Fig. 1e). In the case of the d.c. HDM treatment, more eosinophils and basophils were observed to infiltrate into the dermis of WT mice compared to e.c. HDM treatment (Fig. 1e). However, unlike the e.c. HDM, the d.c. HDM-induced infiltrate of eosinophils and basophils was not abolished in $Tslp^{-/-}$ mice, despite of a partial reduction (Fig. 1e; see Supplementary Fig. 1a for cell counts comparison per microscopic field). We next examined the Th2 cells in the sensitized ears, which are central for allergen sensitization-induced T cell responses. To analyse whether the expression of Th2 cytokines IL-4 and IL-13 is dependent on TSLP, the Il4/Il13 dual-reporter transgenic mice (4C13R$^{Tg}$), in which AmCyan-coding sequence is under IL-4 regulatory elements and DsRed-coding sequence is under IL-13 regulatory elements[15] were bred with $Tslp^{-/-}$ mice to generate $Tslp^{-/-}$ 4C13R$^{Tg}$ mice. As shown in Fig. 1f (see Supplementary Fig. 2a for FACS gating strategy), upon the e.c. HDM treatment, expression of both Amcyan (IL-4) and DsRed (IL-13) was increased in TCR-β$^+$ cells in WT/4C13R$^{Tg}$ ears when compared to e.c. PBS treatment, and such increase was completely abolished in e.c. HDM-treated $Tslp^{-/-}$/4C13R$^{Tg}$ skin (Fig. 1f). Upon d.c. HDM treatment, we observed a higher IL-4 and IL-13 expression by Th2 cells in the skin compared to e.c. HDM (Fig. 1f), and similar to what was observed for eosinophils and basophils, the IL-4 and IL-13 expression by skin Th2 cells was partially reduced but not abolished in $Tslp^{-/-}$ mice (Fig. 1f).

We also examined the Tfh/GC responses in ear-draining lymph nodes (EDLN), which are known to be crucial for allergen sensitization-induced B cell responses and IgE and IgG1 production. In e.c. HDM-treated WT mice, the total LN cell number, Tfh cell (identified as CXCR5$^+$PD1$^+$) frequency in CD4$^+$ T cells and its cell number, as well as IL-4 frequency in Tfh cells and IL-4$^+$ Tfh cell number, were all increased, which were dramatically reduced in e.c. HDM-treated $Tslp^{-/-}$ mice (Fig. 1g–h, see Supplementary Fig. 2b for FACS gating strategy). Correspondingly, the numbers of GC B cells (identified as Fas$^+$GL7$^+$), as well as of IgE$^+$ B cells and IgG1$^+$ B cells (see Supplementary Fig. 2c for FACS gating strategy), were increased in e.c. HDM-treated compared to e.c. PBS-treated WT mice, which were all diminished in $Tslp^{-/-}$ mice (Fig. 1i). This was in agreement with the observation that serum levels of HDM-specific IgE and IgG1 in e.c. HDM-treated $Tslp^{-/-}$ mice were significantly reduced or tended to reduce, respectively (Fig. 1j). In contrast to e.c. HDM, d.c. HDM treatment induced higher numbers for Tfh cells with IL-4 expression and GC B cells in EDLNs of WT mice (Fig. 1g–i). Again, these increases were not abolished in $Tslp^{-/-}$ mice, despite of certain reductions (Fig. 1g–i). Particularly, IgE$^+$ and IgG1$^+$ B cell numbers in EDLNs (Fig. 1i), as well as serum levels of HDM-specific IgE and IgG1 (Fig. 1j), were not reduced in d.c. HDM-treated $Tslp^{-/-}$ mice.

Together, these data indicate that e.c. HDM and d.c. HDM sensitization both induce Th2/Tfh cell responses, however, TSLP is either crucially or only partially implicated in these two contexts.

### TSLP is differentially implicated for e.c. or d.c. HDM sensitization-triggered asthma

We next examined the asthmatic inflammation developed in mice at D13 following i.n. HDM challenge (Fig. 1d). Note that in this experiment, all the mice were i.n. challenged with HDM. We observed first that the e.c. and d.c. PBS treatments (i.e. the vehicle control PBS was applied on LMP_30 μm or LMP_91 μm skin) did not result in any change in bronchoalveolar lavage (BAL) cells (Fig. 2a, b), Th2 cytokine expression (Fig. 2c), eosinophil and basophil infiltration in the lung, or goblet cell hyperplasia (Fig. 2d). Thus, without allergen HDM, skin LMP (at the depth of 30 μm or 91 μm) on its own does not drive any lung

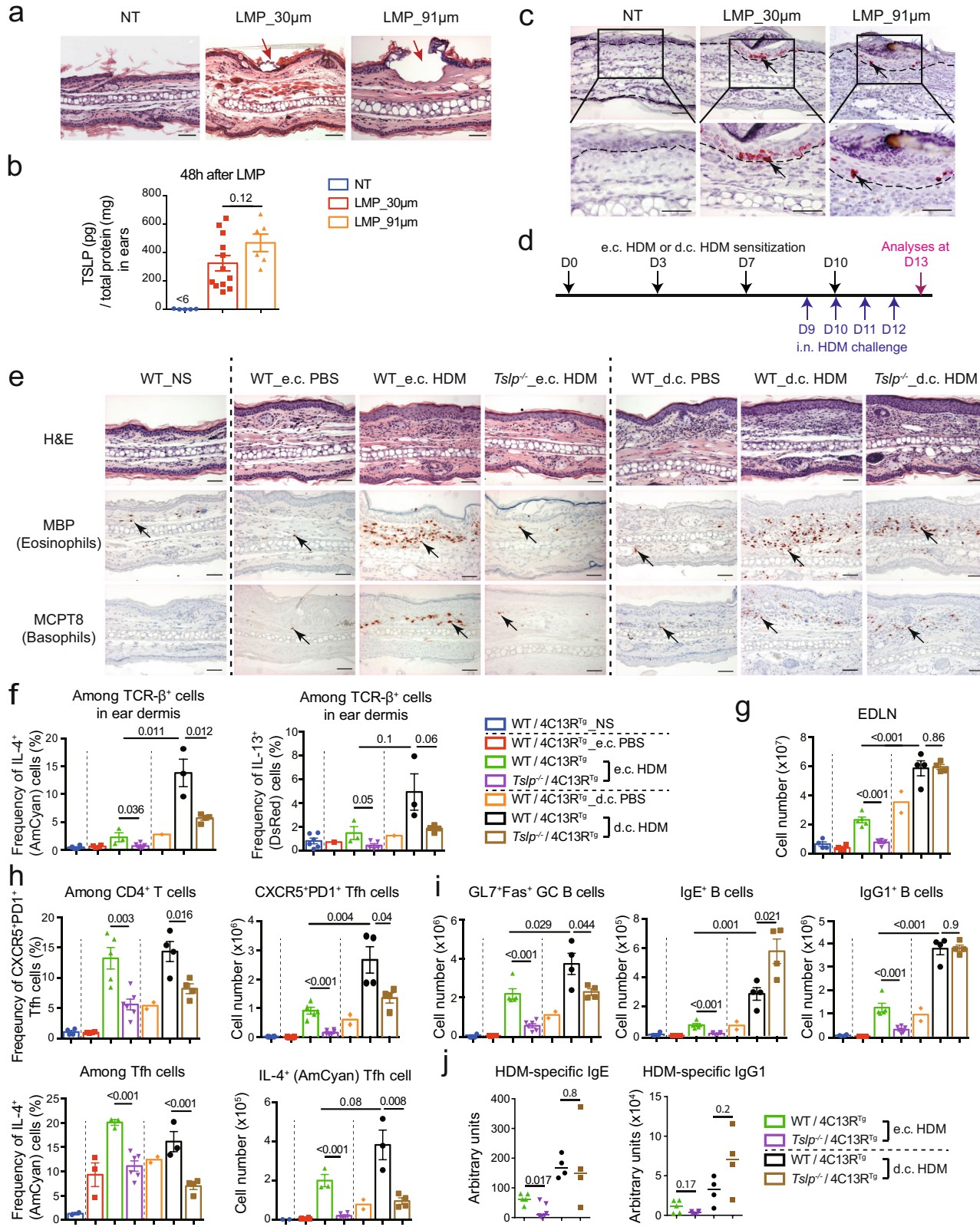

inflammation, indicating that the asthmatic phenotype developed in this experimental protocol is triggered by allergen sensitization through the LMP skin.

Second, BAL cell analyses showed that the number of eosinophils was increased in e.c. HDM-sensitized WT mice compared to e.c. PBS-treated WT mice (Fig. 2b), which was accompanied by an increase of the expression of Th2 cytokines IL-4, IL-5, IL-13 as well as CCR3 (an

indicator for eosinophils) and MCPT8 (an indicator for basophils) by BAL cells (Fig. 2c). All these increases were abolished in $Tslp^{-/-}$ mice (Fig. 2b–c). Moreover, e.c. HDM sensitized WT mice exhibited perivascular and peribronchiolar cell infiltrations, including eosinophils and basophils in the lung (Fig. 2d, hematoxylin & eosin (H&E) and immunohistochemistry (IHC) analyses), mucus-producing goblet cell hyperplasia (Fig. 2d, periodic acid schiff (PAS) staining), as well as

**Fig. 1 | TSLP is differentially implicated for Th2/Tfh responses induced by epicutaneous or dermacutaneous HDM sensitization. a** Hematoxylin & eosin (H&E) stained sections of ears collected immediately after laser microporation (LMP). Red arrow points to a micropore. NT, non-treated. **b** TSLP protein level in ears (*n* = 5, 12, 6 mice). **c** RNAscope in situ hybridization for TSLP mRNA. Black arrows point to one of the positive signals. Dashed lines indicate the dermal/epidermal junction. **d** Experimental protocol. House dust mites (HDM) or PBS were applied on LMP_30 μm ears to realize e.c. sensitization, or on LMP_91 μm ears to realize d.c. sensitization, at day (D) 0, 3, 7 and 10. Mice were challenged intranasally (i.n.) with HDM every day from D9 to D12, and analysed at D13. **e** H&E staining, immunohistochemistry (IHC) staining with anti-MBP antibody (specific for

eosinophils), or anti-MCPT8 antibody (specific for basophils) of ear sections. Black arrows point to one of the positive signals. Scale bar = 50μm (**a**, **c**, **e**). NS, non-sensitized. **f** Frequency of IL-4+ (AmCyan) or IL-13+ (DsRed) cells among CD45+ TCR-β+ cells in ears of mice (*n* = 6, 4, 3, 5, 1, 3, 4 mice). **g** Total cell number in ear-draining lymph nodes (EDLNs). **h** Frequency and cell number of Tfh cells and IL-4+ Tfh cells in EDLNs. **i** Numbers of GL7+Fas+ GC B cells, IgE+ B cells and IgG1+ B cells in EDLNs. For **g**–**i**, *n* = 4, 4, 5, 6, 2, 4, 4 mice. **j** Serum levels of HDM-specific IgE and IgG1 (*n* = 5, 6, 4, 4 mice). Graphs in **b**, **f**–**i** show mean ± SEM, two-sided Student's *t*-test. Graphs in **j** show median, two-sided Mann–Whitney rank sum test. All data are representative of three independent experiments with similar results. Source data are provided as a Source Data file.

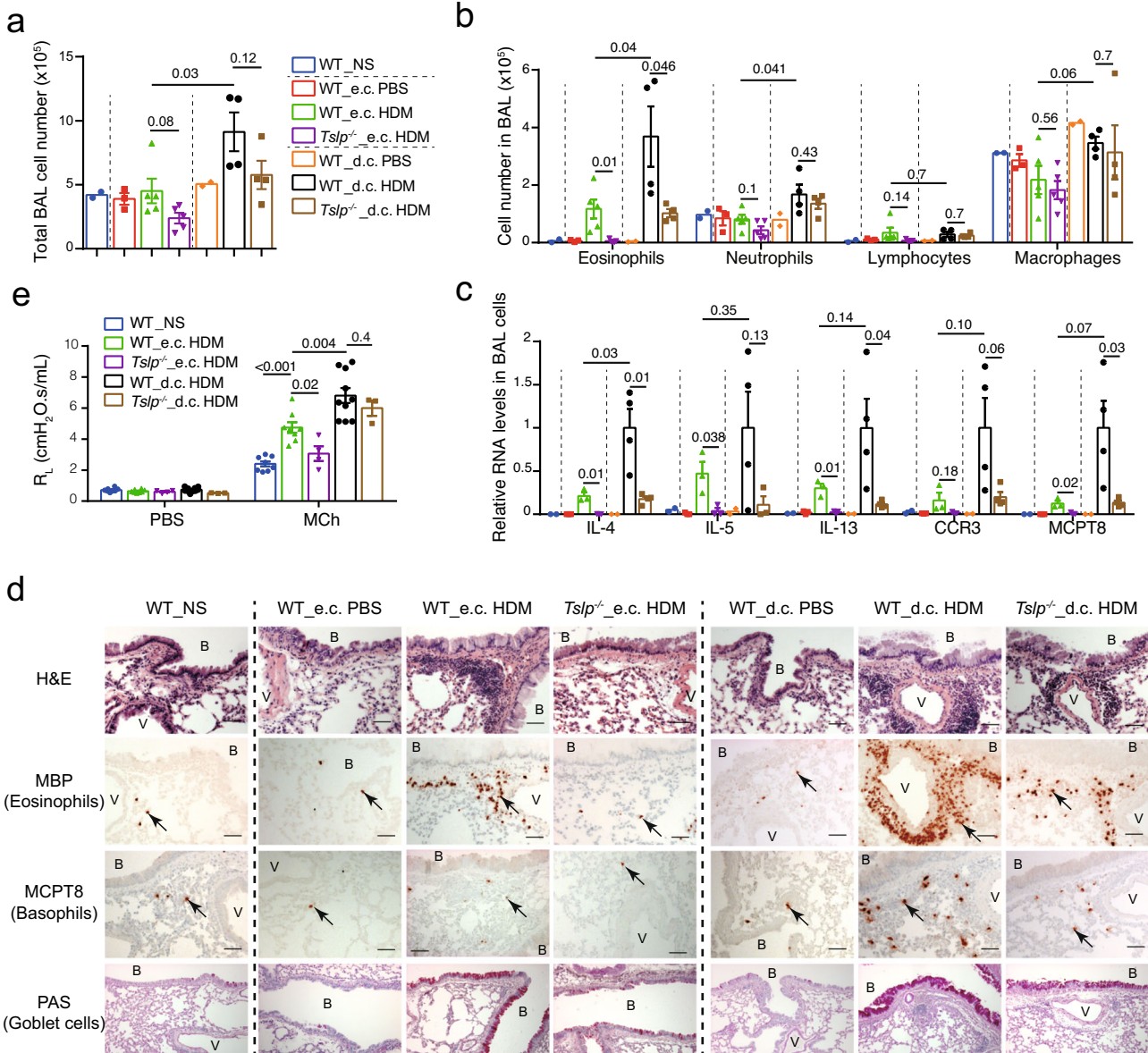

**Fig. 2 | TSLP is differentially implicated for e.c. and d.c. HDM sensitization-induced asthmatic inflammation. a** Total cell number in bronchoalveolar (BAL) fluid. NS, non-sensitized. **b** Cell number of eosinophils, neutrophils, lymphocytes and macrophages in BAL fluid. For **a**, **b**, *n* = 2, 3, 5, 5, 2, 4, 4 mice. **c** Quantitative RT-PCR analyses of BAL cells (*n* = 2, 3, 3, 3, 2, 4, 4 mice). **d** Lung paraffin sections were stained with hematoxylin & eosin (H&E), analysed by immunohistochemistry (IHC) staining with anti-MBP antibody (for eosinophils) or MCPT8 antibody (for basophils), or by Periodic Acid Schiff (PAS) staining for mucus-producing goblet cells

(stained as purple). B: bronchiole. V: blood vessel. Black arrows point to one of the positive cells. Bar = 50μm for all pictures. **e** Lung resistance (R_L) at the baseline (aerosol of PBS) and in response to aerosolized methacholine (Mch; 50 mg/ml), measured by FlexiVent system (*n* = 8, 8, 4, 10, 3 mice). Graphs in **a**–**c**, **e** show mean ± SEM. Two-sided Student's *t*-test. Data are representative of 3 (**a**–**d**) or 2 (**e**) independent experiments with similar results. Source data are provided as a Source Data file.

enhanced airway responsiveness to methacholine (Fig. 2e, shown by lung resistance R$_L$), which were again all abolished in *Tslp*$^{-/-}$ mice. Together, these data indicate that TSLP is crucially required for e.c. HDM-induced allergic asthma.

In contrast to e.c. HDM sensitization, d.c. HDM sensitization appeared to lead to a stronger asthmatic phenotype in WT mice, including a higher number of total cells, including that of eosinophils as well as neutrophils in the BAL (Fig. 2a, b), accompanied by a higher (or a tendency to be higher) RNA level of IL-4, IL-5, IL-13, CCR3 and MCPT8 in BAL cells (Fig. 2c), a stronger infiltration of eosinophils and basophils in the lung (Fig. 2d; see Supplementary Fig. 1b for cell counts comparison), as well as an enhanced goblet cell hyperplasia (Fig. 2d) and airway hyperresponsiveness (AHR) (Fig. 2e). In *Tslp*$^{-/-}$ mice, we observed that some of these d.c. HDM-induced asthmatic phenotypes, including BAL eosinophils, lung infiltration of eosinophils and basophils and goblet cell hyperplasia, were partially reduced however not abolished (Fig. 2a–d), while the AHR was comparable between *Tslp*$^{-/-}$ and WT mice (Fig. 2e).

These results thus indicate that e.c. HDM sensitization promotes a typical Th2 asthmatic inflammation, which is abolished in *Tslp*$^{-/-}$ mice; in contrast, d.c. HDM sensitization-triggered allergic asthma is only partially reduced in *Tslp*$^{-/-}$ mice. Therefore, in addition to TSLP, there should be other factor(s) derived from the skin, which is (are) implicated in d.c. HDM sensitization and thereby impact(s) the subsequent development of allergic asthma.

## IL-1β enhances the e.c. HDM sensitization in a TSLP-independent manner

Searching for what could be the factors in addition to TSLP contributing to the d.c. HDM-induced skin sensitization and subsequent asthma, we found with interest that IL-1β protein level in the skin was much higher in LMP_91 μm ears compared to LMP_30μm ears (Fig. 3b). RNA in situ hybridization analyses showed that while IL-1β-expressing cells were barely detected in non-treated WT skin, some and many more were detected in LMP_30 μm and LMP_91 μm ears, respectively; and different from TSLP, most of IL-1β-expressing cells appeared to be infiltrated immune cells located in the dermis (Fig. 3a; see below for cellular characterization).

To examine whether IL-1β may be functionally responsible for the difference between e.c. HDM- and d.c. HDM-induced sensitization and the subsequent asthma, we designed an experimental protocol in which the recombinant mouse IL-1β was supplemented to e.c. HDM treatment on LMP_30μm ears, followed by HDM i.n. challenge (Fig. 3c). Note that the administration of IL-1β did not result in an increase of TSLP production (by ELISA, Fig. 3d; and by RNAscope analyses, Supplementary Fig. 3). Histology and IHC analyses of ears showed that the co-administration of IL-1β with HDM (e.c. HDM + IL-1β) in WT mice exacerbated the dermal cell infiltration compared to e.c. HDM, including abundant eosinophils and basophils (Fig. 3e; see Supplementary Fig. 1c for cell counts comparison). Although a small increase of eosinophils and basophils was noted in the dermis by IL-1β alone (without HDM treatment), which may suggest an effect of IL-1β in skin inflammation, this was very mild compared to e.c. HDM or e.c. HDM + IL-1β skin (Fig. 3e). Interestingly, we found that in *Tslp*$^{-/-}$ mice, e.c. HDM + IL-1β also induced the infiltration of eosinophils and basophils (Fig. 3e; see Supplementary Fig. 1c for cell counts comparison), as well as Th2 cytokine (particularly IL-13) expression in TCR-β$^+$ cells in the dermis (Supplementary Fig. 4a), indicating that IL-1β is able to promote e.c. HDM-induce skin Th2 inflammation without the need of TSLP.

Analyses of Tfh/GC response in EDLNs showed that e.c. HDM + IL-1β treatment led to an increase in the number of Tfh cells, IL-4-expressing Tfh cells, GC B cells, IgE$^+$ and IgG1$^+$ B cells compared to e.c. HDM treatment, in both WT and *Tslp*$^{-/-}$ mice (Fig. 3f and Supplementary Fig. 4b). Correspondingly, measurement of HDM-specific IgE and IgG1 in sera showed that the co-administration of IL-1β with e.c. HDM enhanced the production of HDM-specific IgE and IgG1 in WT and particularly in *Tslp*$^{-/-}$ mice (Fig. 3g).

We further examined asthmatic phenotypes after the i.n. HDM challenge. We observed first that the e.c. IL-1β alone (without HDM) treatment did not result in any change in BAL cells (Fig. 4a, b), Th2 cytokine expression (Fig. 4c), eosinophil and basophil infiltration in the lung (Fig. 4d, see Supplementary Fig. 1d for cell counts comparison), or goblet cell hyperplasia (Fig. 4d), indicating that without HDM, IL-1β administration in the skin does not promote any asthmatic inflammation on its own, even though e.c. IL-1β slightly enhances the skin inflammation. Second, co-administration of IL-1β in e.c. HDM-treated WT mice led to a strong increase in the number of total BAL cells (Fig. 4a), including that of eosinophils, lymphocytes and neutrophils (Fig. 4b), accompanied by a higher (or a tendency to be higher) expression of IL-4, IL-5, IL-13, CCR3, and MCPT8 in BAL cells (Fig. 4c), a stronger infiltration of eosinophils and basophils, and an increased goblet cell hyperplasia (Fig. 4d). Notably, the exacerbation of all these allergic asthma phenotypes by IL-1β was dependent of T and B cell responses, as no change of lung inflammation was observed in *Rag1*$^{-/-}$ mice (lacking T and B cells) treated with e.c. HDM + IL-1β compared with e.c. HDM alone (Supplementary Fig. 5). Again, the exacerbation of these allergic asthma phenotypes by IL-1β was observed not only in WT mice but also in *Tslp*$^{-/-}$ mice (Fig. 4a–d).

Altogether, these results suggest that co-administration of IL-1β with e.c. HDM sensitization enhances allergen-triggered Th2 and Tfh/GC responses, as well as the subsequent asthmatic inflammation, in a TSLP-independent manner.

## Infiltrated neutrophils and monocytes/macrophages express IL-1β

We further characterized IL-1β-expressing cells in d.c. HDM-sensitized skin. First, flow cytometry analyses (Supplementary Fig. 6, for gating strategy) showed that in d.c. HDM-treated skin, there was a significant infiltration of CD45$^{hi}$ Siglec-F$^-$ CD49b$^-$ Gr-1$^{hi}$ and Gr-1$^{int}$ cells (Fig. 5a), which corresponded to Ly-6G$^+$ Ly-6C$^-$ neutrophils and Ly-6G$^-$ Ly-6C$^+$ monocytes/macrophages, respectively (Supplementary Fig. 6). The frequency of eosinophils (CD45$^+$Siglec-F$^+$SSC-A$^{hi}$), basophils (CD45$^{int}$Siglec-F$^-$CD49b$^+$) or TCR-β$^+$ T cells (CD45$^+$ Siglec-F$^-$ TCR-β$^+$) cells was not significantly increased (Fig. 5a). In contrast, the e.c. HDM treatment induced only a mild infiltration of Gr-1$^{int}$ and Gr-1$^{hi}$ cells (Fig. 5a). Second, intracellular staining of IL-1β showed that the Gr-1$^{int}$ monocytes/macrophages and the Gr-1$^{hi}$ neutrophils recruited to d.c. HDM-treated skin exhibited the highest expression level of IL-1β among the cell populations examined (Fig. 5b). Calculation of frequency of the gated IL-1β$^+$ cells showed that the majority of IL-1β-expressing cells comprised Gr-1$^{hi}$ neutrophils and Gr-1$^{int}$ monocytes/macrophages (Fig. 5c).

We then sought to test whether the depletion of Gr-1$^{hi}$ and/or Gr-1$^{int}$ cells would reduce the IL-1β level in the d.c. HDM-treated skin, by administrating anti-Gr-1 antibody (Ab) (clone NIMP-R14) which depletes both Gr-1$^{hi}$ and Gr-1$^{int}$ cells[16] or anti-Ly6G Ab, which was reported to selectively deplete Gr-1$^{hi}$ cells in Balb/c (but not C57BL/6) WT mice[17,18] (Fig. 5d). As expected, at D1, NIMP-R14 Ab efficiently depleted Gr-1$^{hi}$ neutrophils and Gr-1$^{int}$ monocytes/macrophages, whereas anti-Ly6G Ab depleted only Gr-1$^{hi}$ neutrophils but not Gr-1$^{int}$ monocytes/macrophages, without impacting other cells like T cells or eosinophils (Fig. 5e, f). This was accompanied by a strong reduction of IL-1β level in the d.c. HDM-sensitized skin from the NIMP-R14-injected mice, and to a lesser extent from the anti-Ly6G-injected mice (Fig. 5g). In contrast, TSLP levels remained unchanged (Fig. 5g). These results thus indicate that both Gr-1$^{hi}$ neutrophils and Gr-1$^{int}$ monocytes/macrophages are cellular sources for IL-1β in d.c. HDM-sensitized skin, and that the depletion of these cells reduces IL-1β but does not impact TSLP production.

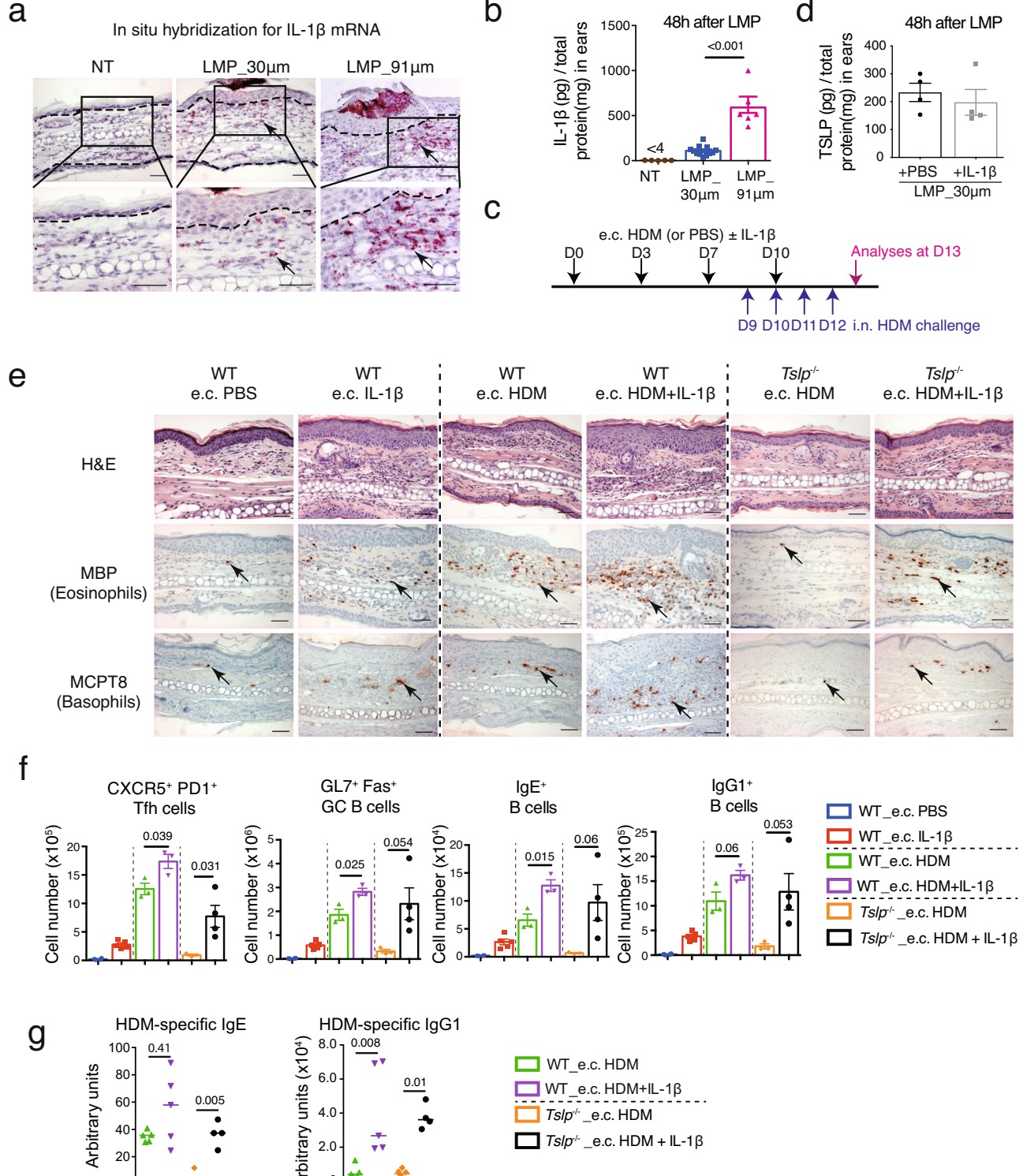

**Fig. 3 | Co-administration of IL-1β enhances e.c. HDM-induced Th2/Tfh responses in a TSLP-independent manner. a** RNAscope in situ hybridization for IL-1β mRNA in non-treated (NT), LMP_30μm and LMP_91 μm ears at 48 h after the microporation. Dashed lines indicate the dermal/epidermal junction. **b** ELISA measurement of IL-1β protein levels in ears at 48 h after LMP_30 μm and 91 μm (*n* = 5, 12, 6 mice). **c** Experimental protocol. HDM with or without IL-1β was applied on LMP_30μm ears (e.c. HDM ± IL-1β), at day (D) 0, D3, D7 and D10. Mice were intranasally (i.n.) challenged with HDM every day from D9 to D12 to induce allergic asthma, and analysed at D13. **d** ELISA measurement of TSLP protein levels in 30 μm-

LMP ears co-administrated with recombinant IL-1β or PBS (*n* = 4 mice). **e** Hematoxylin & eosin (H&E) staining and immunohistochemistry (IHC) staining for MBP or MCPT8 on ear sections. Black arrows point to one of the positive signals. Scale bar = 50 μm for all pictures. **f** Comparison of CXCR5⁺PD1⁺ Tfh cells, GL7⁺Fas⁺ GC B cells, IgE⁺ B and IgG1⁺ B cells in EDLNs (*n* = 4, 5, 3, 3, 4 mice). **g** Serum levels of HDM-specific IgG1 and IgE in HDM-treated mice (*n* = 5, 5, 6, 4 mice). Graphs in **b**, **d**, **f** show mean ± SEM. Two-sided Student's *t*-test. Graphs in **g** show median. Two-sided Mann–Whitney rank sum test. All data are representative of two independent experiments with similar results. Source data are provided as a Source Data file.

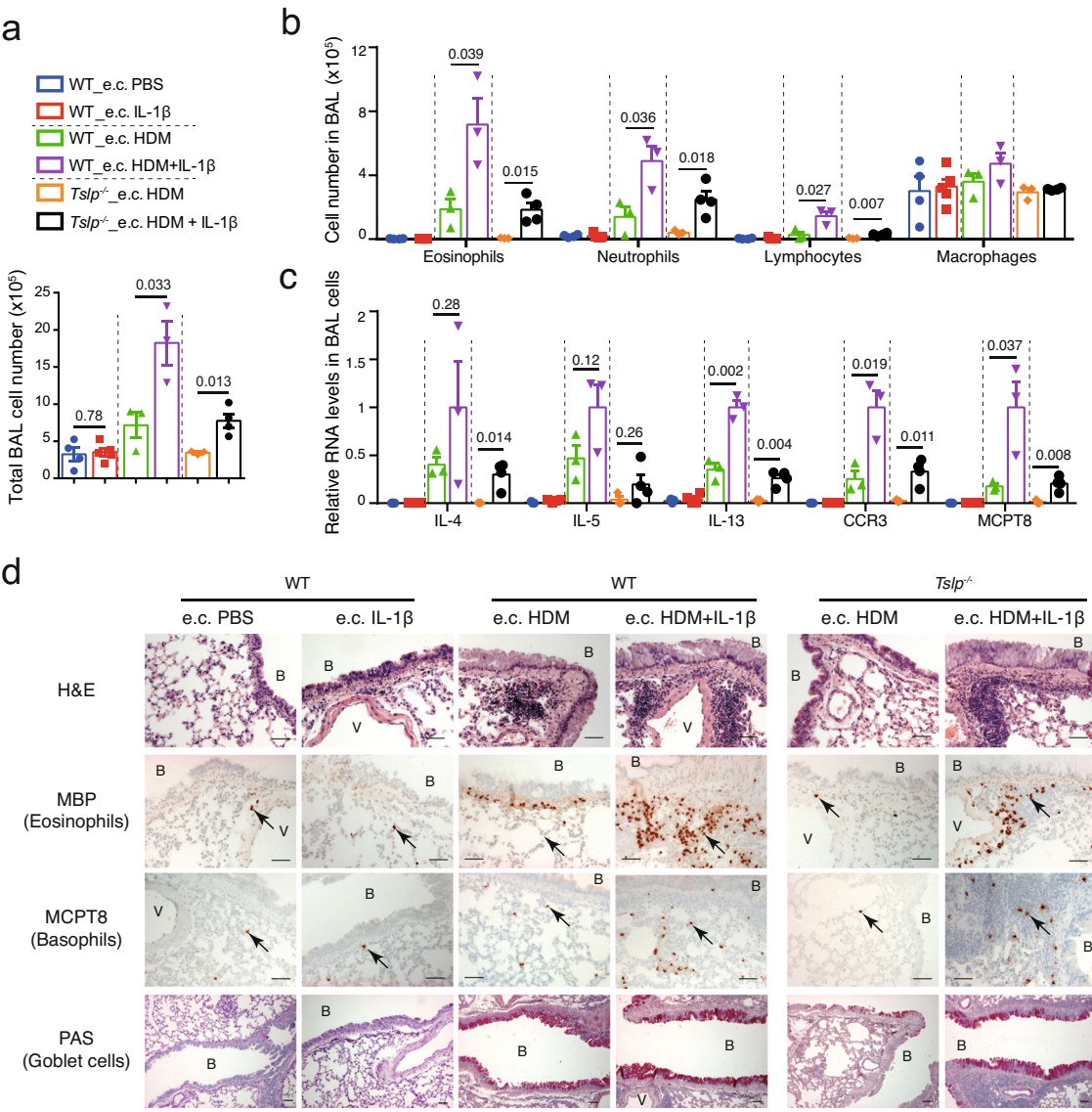

**Fig. 4 | Co-administration of IL-1β exacerbates e.c. HDM-induced asthmatic inflammation in a TSLP-independent manner. a** Total cell number in BAL fluid. **b** Cell number of eosinophils, neutrophils, lymphocytes and macrophages in BAL fluid. For **a, b** $n = 4, 5, 3, 3, 3, 4$ mice. **c** Quantitative RT-PCR analyses of BAL cells ($n = 3, 4, 3, 3, 3, 4$ mice). Graphs in **a–c** show mean ± SEM. Two-sided Student's *t*-test. **d** Lung sections were stained with hematoxylin & eosin (H&E), analysed by immunohistochemistry (IHC) staining for MBP or MCPT8 (stained as dark red, pointed by black arrows), or by Periodic Acid Schiff (PAS) staining (stained as purple). B: bronchiole. V: blood vessel. Scale bar = 50 µm for all pictures. All data are representative of three independent experiments with similar results. Source data are provided as a Source Data file.

We also examined whether the infiltration of IL-1β-expressing neutrophils and monocytes/macrophages in the skin requires TSLP. Upon e.c. HDM or d.c. HDM treatment, WT and *Tslp*[−/−] mice exhibited similar levels for IL-1β (Fig. 5h) and similar frequencies for Gr-1[hi] and Gr-1[int] cells in the skin (Fig. 5i). In addition, administration of recombinant TSLP did not induce IL-1β level in e.c. HDM-treated skin (Fig. 5j). These results thus indicate that the infiltration of IL-1β-expressing neutrophils and monocytes/macrophages is an event independent of TSLP signalling.

Moreover, we observed that the increased infiltration of neutrophils and monocytes/macrophages in d.c. HDM compared to e.c. HDM treatment was associated with the higher induction of neutrophil-chemoattractant factors[19,20] including CXCL2, CXCL3, CXCL5, CCL3, S100A7, S100A8 and S100A9 (but not CXCL1, CCL2 or IL-17C) in the skin (Supplementary Fig. 7a). Notably, their expression in the epidermis exhibited a higher level in d.c. HDM-treated compared to e.c. HDM-treated mice (Supplementary Fig. 7b), suggesting that these

chemoattractant factors, which are possibly derived (or at least partially) from the epidermis, could be implicated in mediating the infiltration of IL-1β-expressing neutrophils and monocyte/macrophages.

## NIMP-R14 antibody treatment during d.c. HDM sensitization reduces the subsequent asthma

We then wondered whether the depletion of the IL-1β-producing cells during the d.c. HDM-sensitization phase (but not during the i.n. challenge) led to the reduction of the subsequent allergic asthma. To test that, mice were i.p. injected at D-1 and D2 with NIMP-R14 or anti-Ly6G Ab, and d.c. sensitized with HDM at D0 and D3, followed by i.n. challenge with HDM at D10-13 (Fig. 6a). This experimental protocol was designed based on the previous report that Gr-1[hi] and Gr-1[int] cells could be efficiently depleted 1 day after the i.p. injection of NIMP-R14 Ab, but started to recover 4 days after[21]. We confirmed that a repeated Ab injection at D2 maintained the cell depletion during the d.c. sensitization phase, while Gr-1[hi] and Gr-1[int] cells were recovered before the i.n.

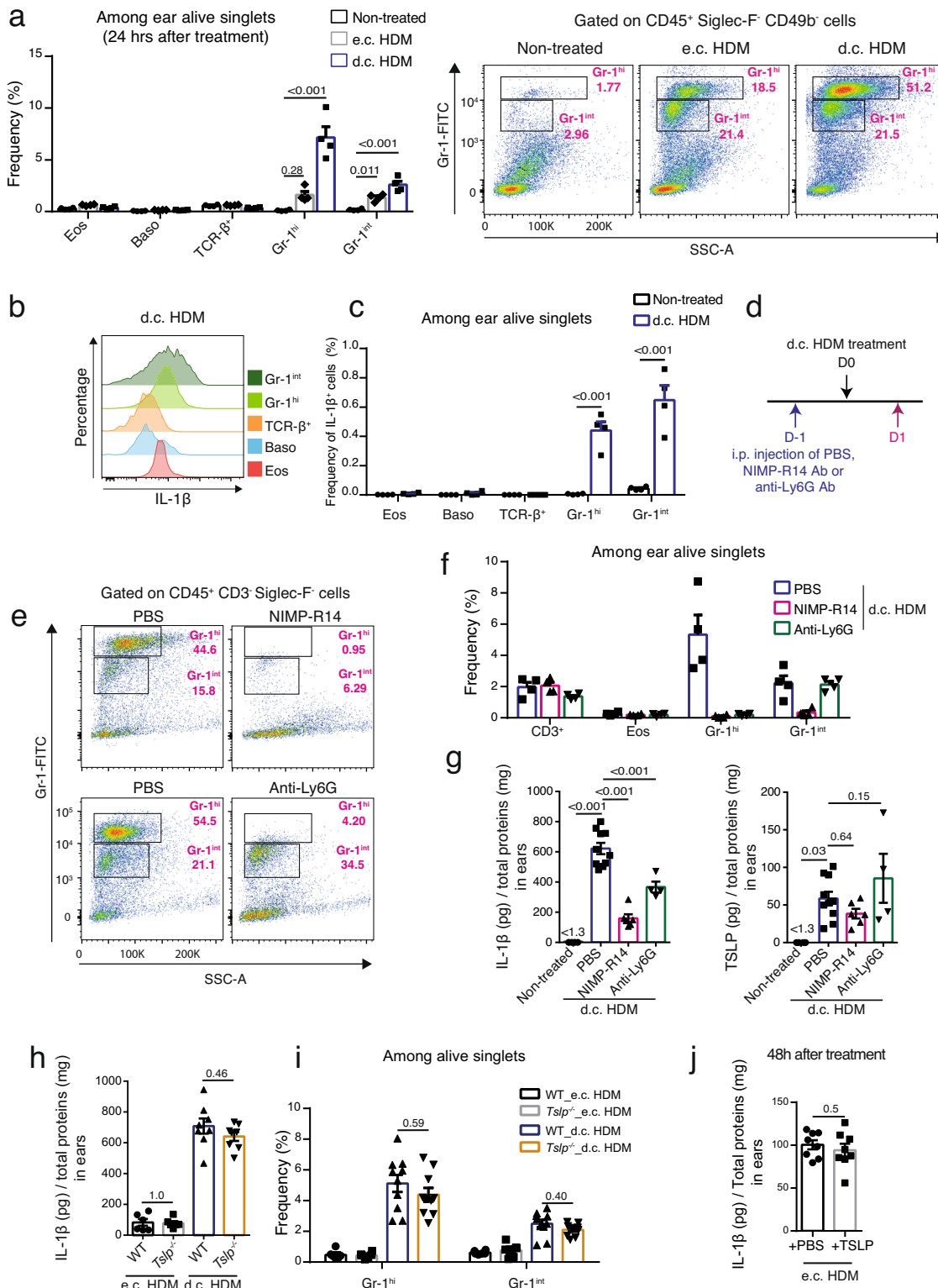

HDM challenge (at D9, Supplementary Fig. 8), therefore allowing us to investigate the role of these cells in the sensitization phase.

Analyses of BAL at D14 following i.n. HDM challenge showed that total cell number was decreased in both NIMP-R14- and anti-Ly6G-injected mice (Fig. 6b). Cell differential counting showed that eosinophil numbers were decreased in both NIMP-R14- and anti-Ly6G-injected mice compared to PBS-treated ones, whereas lymphocyte cell number was significantly decreased in NIMP-R14- but not in Ly6G-injected mice (Fig. 6b). In contrast, neutrophil number was

comparable between PBS-, NIMP-R14- and anti-Ly6G-injected mice (Fig. 6b). RT-qPCR analyses of BAL cells showed a decrease in RNA levels of IL-13, IL-5, IL-4, CCR3 and MCPT8 in both NIMP-R14- and anti-Ly6G-injected mice, although it appeared more prominent with NIMP-R14 than anti-Ly6G (Fig. 6c). In keeping with these data, lung histological analyses showed that NIMP-R14 or anti-Ly6G-injected mice exhibited a reduced peribronchiolar and perivascular infiltration (H&E, Fig. 6d), as well as a reduced hyperplasia of mucus-secreting goblet cells (PAS staining, Fig. 6d). Finally, ELISA analyses showed that HDM-

**Fig. 5 | The d.c. HDM treatment induces the infiltration of IL-1β-expressing Gr-1ʰⁱ and Gr-1ⁱⁿᵗ cells in the skin. a–c** Analyses of IL-1β-expressing cells in the skin. Wildtype Balb/c mouse ears were subjected to d.c. or e.c. HDM treatment and were analysed 24 h later. **a** Left, frequencies of eosinophils (Eos), basophils (Baso), TCR-β⁺ T, Gr-1ʰⁱ and Gr-1ⁱⁿᵗ cells among alive singlets. Right, representative FACS plots. *n* = 4 mice. **b** Histogram comparison of IL-1β in cells from the d.c. HDM-treated ears. **c** Frequencies of IL-1β⁺ cells among alive singlets (*n* = 4 mice). **d–g** Depletion of neutrophils and monocytes/macrophages reduces IL-1β in d.c. HDM-treated ears. **d** Experimental protocol. Wildtype Balb/c mice were intraperitoneally (i.p.) injected with PBS, NIMP-R14 (anti-Gr-1) or anti-Ly6G antibody (Ab) at day (D) -1. Ears were d.c. HDM-treated at D0 and analysed at D1. **e** Representative

FACS plots showing the depletion of both Gr-1ʰⁱ and Gr-1ⁱⁿᵗ cells by NIMP-R14 Ab, or of Gr-1ʰⁱ cells by anti-Ly6G Ab. **f** Frequencies of cells among ear alive singlets (*n* = 4 mice). **g** IL-1β and TSLP protein levels in ears (*n* = 4, 10, 6, 4 mice). **h–j** Infiltration of IL-1β-expressing cells is TSLP-independent. Ears of WT or *Tslp⁻/⁻* mice were treated with e.c. HDM or d.c. HDM at D0 and analysed at D1 for IL-1β (**h**) and for Gr-1ʰⁱ and Gr-1ⁱⁿᵗ cells (**i**). For **h**, *n* = 6, 6, 8, 8 mice. For **i**, *n* = 6, 6, 10, 10 mice. **j** Recombinant TSLP or PBS were administrated on e.c. HDM-sensitized ears and IL-1β levels were measured (*n* = 8 mice). Graphs in **a, c, f–j** show mean ± SEM. **a, c, f–i** One-way ANOVA test; **c, j** Two-sided Student's *t* test. Data are representative of 3 (**a–c, e–i**) or 1 (**j**) independent experiments with similar results. Source data are provided as a Source Data file.

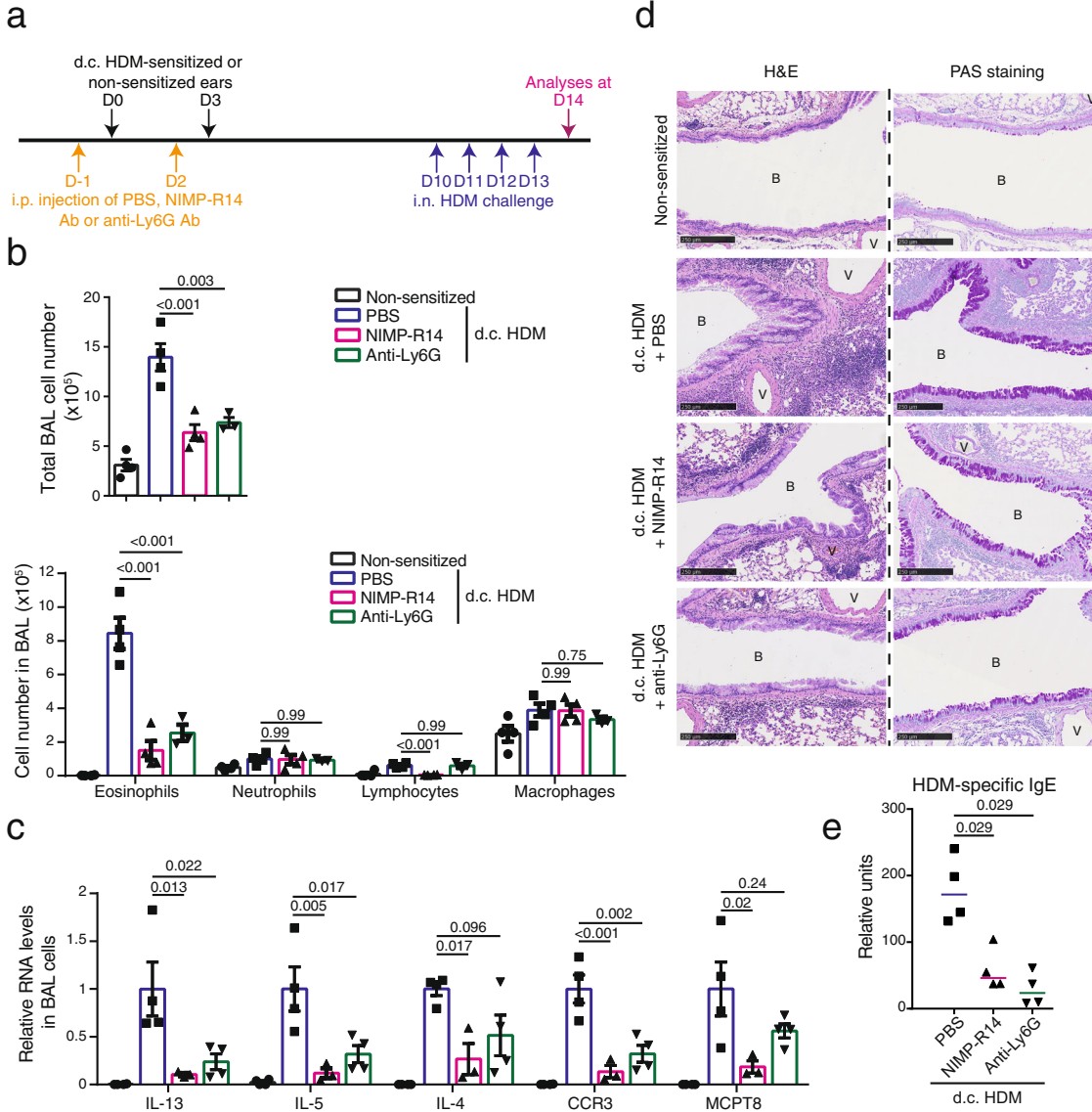

**Fig. 6 | Depletion of IL-1β-expressing cells during d.c. HDM sensitization reduces the subsequent asthmatic phenotype. a** Experimental protocol. Wild-type Balb/c mice were intraperitoneally (i.p.) injected with PBS, NIMP-R14 or anti-Ly6G antibody (Ab) at day (D) -1 and D2. Mice were d.c. sensitized with HDM on LMP_91μm ears at D0 and D3 or non-sensitized. All mice were intranasally (i.n.) challenged with HDM from D10 to D13 and analysed at D14. **b** Total cell number and differential cell counting in BAL fluid (*n* = 4, 4, 4, 3 mice). **c** Relative mRNA levels of

genes in BAL cells (*n* = 4, 4, 3, 4 mice). **d** Lung sections were stained with hematoxylin-eosin (H&E) for histological analyses or Periodic Acid Schiff (PAS) for goblet cell hyperplasia analyses. B: bronchiole. V: blood vessel. Bar = 250 μm for all pictures. **e** Serum level of HDM-specific IgE measured by ELISA (*n* = 3 mice). Graphs in **b, c** show mean ± SEM, One-way ANOVA test. Graph in **e** marks median, two-sided Mann–Whitney rank sum test. All data are representative of two independent experiments with similar result. Source data are provided as a Source Data file.

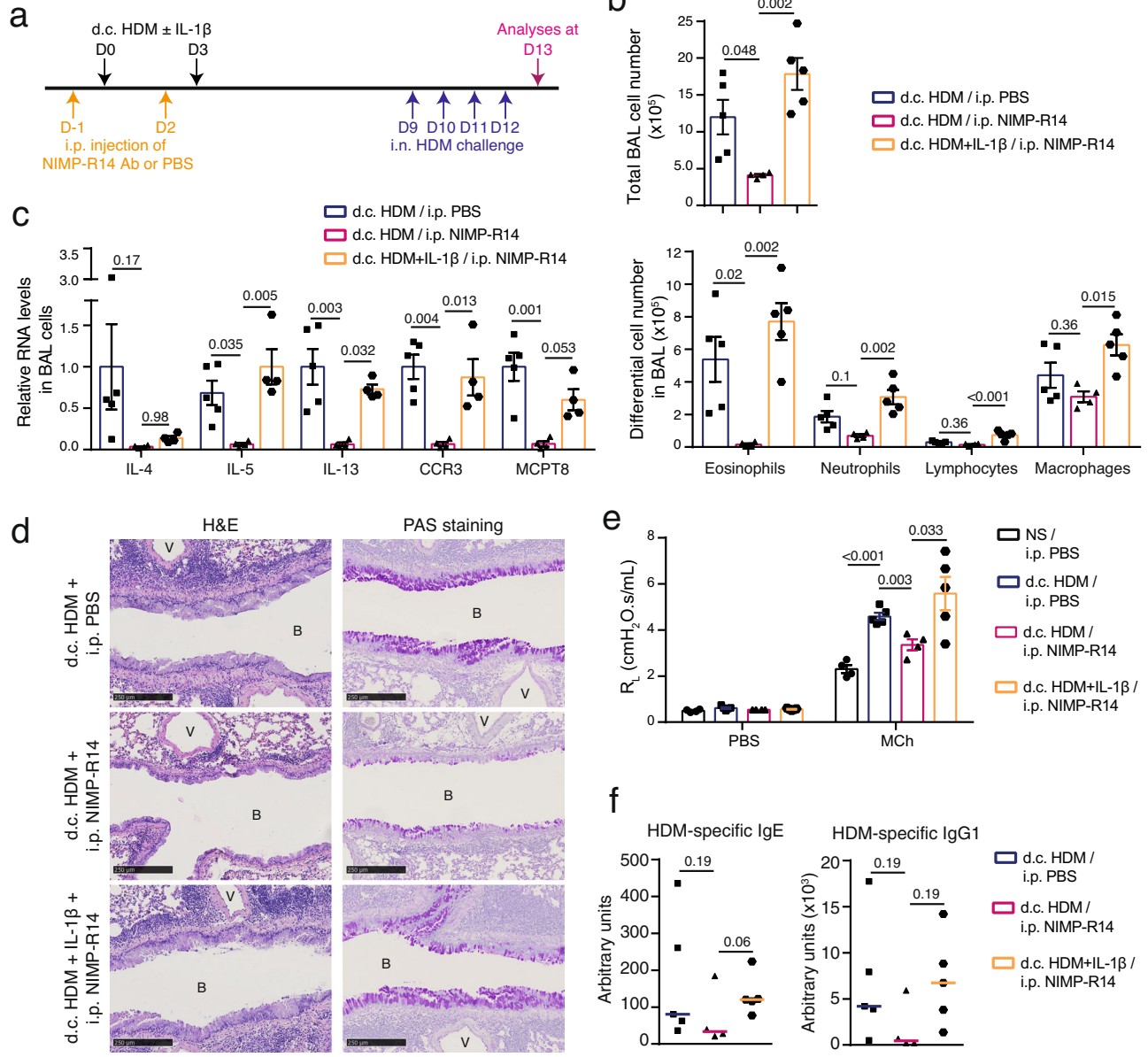

**Fig. 7 | Administration of IL-1β restores the d.c. HDM-induced asthmatic phenotype in NIMP-R14-treated mice. a** Experimental protocol. Wildtype Balb/c mice were intraperitoneally (i.p.) injected with PBS or NIMP-R14 antibody at day (D) -1 and D2. Mice were d.c. sensitized with HDM ± IL-1β on LMP_91μm ears at D0 and D3. All mice were intranasally (i.n.) challenged with HDM from D9 to D12 and analysed at D13. **b** Total cell number and differential cell counting in BAL fluid (*n* = 5, 4, 5 mice). One-way ANOVA test. **c** Relative RNA levels of genes in BAL cells (*n* = 5, 4, 4 mice). One-way ANOVA test. **d** Lung sections were stained with hematoxylin-eosin (H&E) or Periodic Acid Schiff (PAS). B: bronchiole. V: blood vessel. Bar = 250 μm for all pictures. **e** Lung resistance ($R_L$) at the baseline (aerosol of PBS) and in response to aerosolized methacholine (Mch; 50 mg/ml), measured by FlexiVent system (*n* = 4, 5, 4, 5 mice). Two-sided Student's *t* test. **f** Serum level of HDM-specific IgE and IgG1 measured by ELISA (*n* = 5, 4, 5 mice). Two-sided Mann–Whitney rank sum test. NS, non-sensitized. Data are representative of 2 (**b**–**d**, **f**) or 1 (**e**) independent experiments with similar results. Source data are provided as a Source Data file.

specific IgE was significantly reduced in both NIMP-R14 and anti-Ly6G-treated mice (Fig. 6e).

Together, these results indicate that the depletion of Gr-1[hi] and Gr-1[int] cells, or the depletion of Gr-1[hi] cells alone, during the d.c. HDM sensitization, reduces the subsequent lung inflammation, suggesting that these IL-1β-expressing cells, particularly neutrophils, are crucial for d.c. HDM sensitization-triggered allergic asthma.

### Skin IL-1β restores asthmatic inflammation in NIMP-R14-treated mice

We further asked whether the role of Gr-1[hi] and Gr-1[int] cells in d.c. HDM sensitization-triggered allergic asthma is mediated through IL-1β. To answer this question, Balb/c WT mice were i.p. injected with NIMP-R14 Ab at D-1 and D2, and d.c. sensitized with HDM supplemented with IL-1β at D0 and D3, followed by i.n. HDM challenge at D9-12 (Fig. 7a). Analyses at D13 showed that the co-administration of IL-1β largely restored the allergic asthmatic phenotypes in NIMP-R14-injected mice, from the number of total BAL cells including that of eosinophils and lymphocytes (Fig. 7b), the expression of IL-5, IL-13, CCR3, MCPT8 (and to a lesser extent, IL-4) by BAL cells (Fig. 7c), to the perivascular and peribronchiolar cell infiltration and goblet cell hyperplasia (Fig. 7d). In addition, measurement of lung resistance showed that the d.c. HDM-induced AHR was diminished in NIMP-R14-injected mice, which was restored by the co-administration of IL-1β (Fig. 7e). Finally, serum HDM-specific IgE and IgG1 levels also tended to be restored by IL-1β in the NIMP-R14-injected mice (Fig. 7f). Together, these results suggest

that IL-1β mediates the role of Gr-1[hi] and/or Gr-1[int] cells for d.c. HDM sensitization-triggered allergic asthma.

Moreover, we tested whether a direct blockade of IL-1β signalling, by administrating anti-IL-1β Ab or Anakinra (a recombinant IL-1 receptor antagonist) during the d.c. HDM sensitization, could reduce the subsequent asthmatic phenotype. Although the effects observed were less striking compared to the depletion with NIMP-R14 Ab, the results showed that mice with i.p. injection with anti-IL-1β during d.c. HDM sensitization developed a weaker asthmatic inflammation, including a decreased number of eosinophils in BAL cells, a decrease tendency for RNA levels of IL-13, IL-5, CCR3 and MCPT8 (but not IL-4), a milder and more patchy hyperplasia of goblet cells (Supplementary Fig. 9 a–d). We noted that the H&E staining did not show a striking reduction for inflammatory cell infiltration in the lung (Supplementary Fig. 9d). Injection of Anakinra did not reach a better reduction for lung inflammation either (Supplementary Fig. 9e–f), suggesting that unlike the depletion of IL-1β-expressing neutrophils and monocytes/macrophages, the blockade of IL-1β or IL-1 signalling using the available anti-IL-1β Ab or anakinra only reaches a mild effect in reducing the d.c. HDM sensitization-triggered allergic asthma.

## Discussion

In this study, we have modelled in mice the allergen sensitization occurring in the skin with barrier disruption at different anatomic depth, either superficially (named epicutaneous e.c. sensitization) or deeply (named dermacutaneous d.c. sensitization). We report that compared with e.c. HDM sensitization, d.c. HDM sensitization leads to stronger Th2 and Tfh/GC cell responses accompanied by type 2 skin inflammation, and consequently, it triggers a stronger allergic lung inflammation as well as AHR upon the i.n. challenge of HDM. We provide experimental evidence that in these two contexts, epidermal keratinocyte-derived TSLP plays either a major or only a partial role for HDM skin sensitization and the subsequent allergic asthma. We further identify that IL-1β, whose level is induced by a deeper barrier disruption and whose expression is detected in neutrophils and monocytes/macrophages infiltrated to the skin, promotes the d.c. HDM sensitization and the atopic march in a TSLP-independent manner. Together, our study suggests context-dependent roles for TSLP and IL-1β in skin allergic sensitization and atopic march (Fig. 8): for e.c. sensitization occurring more superficially in the skin, which may correspond to the context of milder AD, TSLP is dominantly crucial, while for d.c. sensitization occurring more deeply in the skin, which may correspond to the context of more severe AD, IL-1β is another important player, in addition to TSLP, to generate the allergen sensitization and develop the atopic march.

It has been reported that TSLP overexpressed by epidermal keratinocytes in AD lesions[22] drives AD pathogenesis[23–26] and the atopic march[8,27–29], with its role recognized in promoting Th2 and Tfh cell differentiation in mouse and human[13,30–32]. The current study provides further evidence on the context-dependent contribution of TSLP in allergen sensitization which occurs at different depth of the skin mimicking the heterogeneous situations in AD lesion. Note that first, TSLP is similarly induced by superficial or deep disruption of the barrier skin, and second, its functional contribution to allergic sensitization and atopic march decreases once skin sensitization goes deeper. This finding may thus provide insight for the recent clinical trial results in AD obtained from TSLP neutralizing antibody Tezepelumab, showing that moderate to severe AD adults treated with Tezepelumab presented only a numeric but not significant improvement compared to placebo-treated group[33]. Very recently, a phase 2b clinical trial (NCT03809663) was terminated/withdrawn because Tezepelumab as a monotherapy in moderate to severe AD patients did not reach the efficacy required. As these clinical trials with limited/negative results were performed in moderate to severe AD patients, the lack of efficacy of Tezepelumab could be at least partially explained by the context-

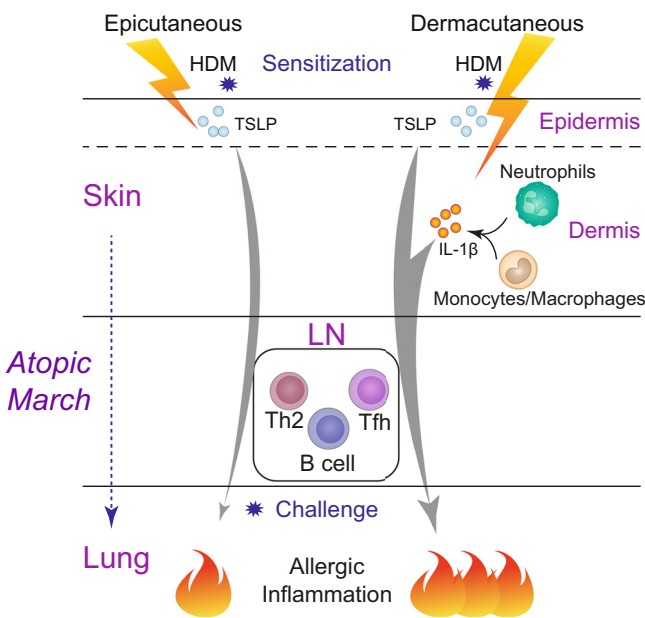

**Fig. 8 | A schematic representation of the context-dependent role of TSLP and IL-1β in promoting skin allergic sensitization and the atopic march.** When allergen HDM sensitization occurs superficially in the skin (epicutaneous sensitization), TSLP derived from keratinocytes located in the epidermis plays a dominantly crucial role for allergic sensitization through the lymph node (LN; generating Th2, Tfh and B cell responses) and the subsequent allergic inflammation in the lung. When allergen sensitization occurs deeply in the skin (dermacutaneous sensitization), IL-1β derived from the infiltrated neutrophils and monocytes/macrophages contributes together with TSLP, to generate a stronger allergen sensitization and subsequently a more severe lung allergic inflammation.

dependent implication of TSLP in allergic skin inflammation and sensitization in these patients.

We show in this study that IL-1β level is elevated following a deeper disruption of the skin, but unlike TSLP produced by epidermal keratinocytes, infiltrated Gr-1[hi] (Ly-6G[+]) neutrophils and Gr-1[int] (Ly-6C[+]) monocytes/macrophages are the major cellular sources for skin IL-1β. Actually, elevated IL-1β level has been reported not only in the skin of AD-like mouse models with genetic mutations of *Flg*[34], *Cdsn*[35], or *Spink5*[36], but also in human AD lesional skin[37,38]. Moreover, serum IL-1β levels were shown to be significantly increased in AD patients[39], to correlate with the severity of AD[40], and following improvement to decrease in a significant manner[40]. Our study provides experimental evidence for the role of IL-1β in skin sensitization and the atopic march: first, the supplement of IL-1β exacerbates e.c. HDM-sensitization (where endogenous IL-1β level is low) and the subsequent asthma, thus somehow mimicking the effects of d.c. HDM sensitization; second, the depletion of IL-1β-expressing Gr-1[hi] or Gr-1[hi+int] cells reduces the d.c. HDM-triggered asthma, which is restored by the skin administration of IL-1β. One limitation of our study is that the blockade with anti-IL-1β or Anakinra (an IL-1 receptor antagonist, IL-1RA) appears to reach only limited reduction in asthmatic phenotypes. We suspect that this could be due to the inefficiency of the blockade. For example, Anakinra blockade efficiency in mice seems to be influenced by genetic background[41], and its treatment has led to both positive and negative results in different human diseases suggesting complex role of IL-1 receptor antagonist and its usage difficulties from tissues to tissues. The ongoing efforts for developing more potent treatments than Anakinra[42] may hopefully provide better reagents for targeting IL-1 signalling.

Importantly, we conclude from our study that IL-1β and TSLP are two independent factors promoting allergic sensitization and atopic

march. In one hand, we show that first, the depletion of IL-1β-expressing neutrophils and monocytes/macrophages does not impact TSLP production; second, the supplement of IL-1β does not induce TSLP expression, which is in contrast to a previous report showing that in vitro IL-1β induces TSLP in reconstructed human epidermis culture[43]; and third, the supplement of IL-1β enhances the e.c. HDM sensitization in $Tslp^{-/-}$ mice similarly as in WT mice, suggesting that the effect of IL-1β in promoting skin allergic sensitization is TSLP-independent. On the other hand, we show that the supplement of TSLP does not increase IL-1β level in the skin, and that TSLP is not required for d.c. HDM-induced IL-1β or infiltration of neutrophils and monocytes/macrophages, in keeping with a recent paper reporting TSLP-TSLPR signalling is not implicated in skin recruitment of neutrophils[44]. Taken together, these data emerge IL-1β as a potential target which is independent of TSLP for allergic sensitization in AD patients, particularly those moderate to severe cases.

As a plurifunctional cytokine, IL-1β is known to be implicated in innate and adaptive immunity. Here we report an adjuvant function of IL-1β in promoting allergen-triggered skin sensitization and asthma, rather than an immune regulation function on its own. Also, IL-1β appears to act rather locally than systemically, as an increase in IL-1β protein was detected in the skin, but not in a distant organ like lung or in circulation at any time points examined upon d.c. HDM treatment (Supplementary Fig. 10). In particular, we show that skin IL-1β promotes allergen-induced Th2 and Tfh cell differentiation in the skin-draining LNs, both of which are critically implicated in T- and B-cell mediated memory to allergen during the atopic march process. In agreement with our data, the role of IL-1β on Th2 response has been also reported in several studies of allergy mouse models, where lung inflammation and/or Th2 cytokines are decreased in $Il1b^{-/-}$ mice sensitized by i.p. OVA/alum and i.n. challenged by OVA[45], or exacerbated in mice co-treated with IL-1β with i.n. OVA[46,47]. Besides, it has been also shown that IL-1β promotes the differentiation of Tfh cells, which express IL-4 and promote GC B cells and antigen-specific IgE and IgG1, upon i.p. immunization with OVA/Alum[48] or i.n. OVA or peanut exposure[47,49]. Despite all these studies, cellular and molecular mechanisms underlying the role of IL-1β for Th2/Tfh cell differentiation remain still to be investigated. It is possible that IL-1β activates dendritic cells (DCs) as previously shown in mouse and human[50,51], or directly acts on T cells[52]. For example, Ritvo et al. showed that Tfh cells expressed IL-1β receptor IL-1R1, and in vitro stimulation of Tfh cells with IL-1β induces their production of IL-4[48]. Recently, IL-1β was also reported to act on CD4$^+$ T cells to induce Bcl6, CXCR5 and ICOS expression thus promoting Tfh cells in response to live vaccines[53].

In addition to its prominent effect on enhancing eosinophils and basophils in lung inflammation developed following the i.n. HDM challenge, we note that skin IL-1β has also an impact, although less striking, in the elevation of neutrophils in BAL. First, compared to e.c. HDM sensitization, d.c. HDM sensitization triggers a small but significant increase in neutrophils in BAL (Fig. 2b). Second, we show that co-delivery of IL-1β during e.c. HDM sensitization leads to a higher neutrophil number in BAL (Fig. 4b). Third, the administration of IL-1β appears to restore the BAL neutrophil number in d.c. HDM-sensitized WT mice depleted of Gr-1 cells (Fig. 7b). In contrast to IL-1β, TSLP does not seem to have any role in BAL neutrophils, as their number remains unchanged between d.c. HDM-sensitized WT and $Tslp^{-/-}$ mice (Fig. 2b). Given the recognized role of airway neutrophils in persistent and severe asthma associated with corticosteroid-resistance[54], it will be interesting to further investigate in mice whether allergic asthma developed following d.c. sensitization is more resistant to corticosteroid, and to explore in patients the possible link of skin IL-1β in AD lesions with the development of corticosteroid-resistant asthma with neutrophilic inflammation.

Our data point to a role for neutrophils for the development of skin allergic sensitization and the atopic march. These cells are early recruited in the skin, and their numbers are positively correlated with the depth of barrier disruption and the expression of chemoattractant factors in the skin. Notably, neutrophils are recognized to for their importance in healing tissue injury[55] and in host defence against microbial pathogens including S. aureus[56]. As AD skin is commonly colonized with S. aureus, one may expect that neutrophils and their derived IL-1β act as double-sided sword, on the one hand contributing to S. aureus clearance, and on the other hand, as shown in this study, contributing to the promotion and exacerbation of skin allergic sensitization, which provides one plausible explanation for the association of S. aureus in AD with the development of atopic march[57]. In keeping with our data, an early recruitment of neutrophils was shown to promote contact allergic sensitization of hapten[21]. The role of neutrophils in airway allergic sensitization was also reported, showing that recruitment of neutrophils to the lung promotes ragweed pollen extract allergic airway inflammation[58]. Interestingly, neutrophils were found to mediate the recruitment of DCs to the site of L. major infection[59], for DCs migration to the draining LNs in contact dermatitis[21], and for skewing T cells towards a Th2 phenotype in a model of L. major infection[60]. Whether this is the case in our model remains to be determined. Finally, even though our study has suggested that IL-1β is one key factor mediating the promoting effect of neutrophils in allergic sensitization and the subsequent allergic asthma, it is possible that other factors derived from neutrophils including neutrophil extracellular traps[61] may also have a contribution.

To conclude, our study points to the importance of different microenvironmental factors that are induced by the barrier disruption at different anatomic depth in regulating the allergen sensitization through skin. It reveals that TSLP and IL-1β, produced by different cellular compartments, represent two important players in promoting skin allergen sensitization and atopic march in a context-dependent manner. Our data highlight the importance towards precision medicine in order to achieve better therapeutic/preventative efficiency in individual AD patients. In addition, it points to the necessity to consider and to further test the combined treatment, for example, the blockade of TSLP (Tezepelumab) and IL-1β signalling (Canakinumab, Anakinra or new developed reagents) in moderate to severe AD, which could be beneficial not only for AD inflammation, but also for allergic sensitization, thus preventing or reducing the risk of the atopic march. Certainly, given their multiple functions, for example, IL-1β in host defence against microbial pathogens, careful design of targeting strategies including administration routes and evaluation of benefits will be crucially required.

## Methods
### Mice
Breeding and maintenance of mice were performed under institutional guidelines, and all experimental protocols were approved by the animal care and ethics committee of animal experimentation of the IGBMC n°017 and by the Ministère de l'enseignement supérieur, de la recherche et de l'innovation. Balb/c and $Rag1^{-/-}$ mice (strain #. 002216) were purchased from the Jackson Laboratory. $Tslp^{-/-}$ mice were previously described[23]. 4C13R dual reporter mice[15] were kindly provided by Dr. W.E. Paul (NIH, USA). All the mouse lines used in the study were backcrossed to Balb/c background (>10 generations). 4C13R$^{Tg}$ reporter mice were bred with $Tslp^{-/-}$ mice to generate $Tslp^{-/-}$ 4C13R$^{Tg}$ mice (in Balb/c background). Ten to fifteen weeks old female mice were used in all experiments. Mice were housed at a temperature of 22 °C, humidity of 40–60% in a 12 h light/12 h dark cycle, with unlimited access to food and water.

### HDM cutaneous sensitization and airway challenge
P.L.E.A.S.E portable (Pantec Biosolutions) laser-assisted skin microporation (LMP) was performed on the dorsal side of mouse ears. Two sets of parameters were optimized. For the depth of 30 μm

(LMP_30μm): 2 pulses per pore, with fluence of 7.5 J/cm$^2$, pulse length of 75 μs, RepRate of 500 Hz and power of 1.0 W; for the depth of 91 μm (LMP_91 μm): 2 pulses per pore with fluence of 22.7 J/cm$^2$, pulse length of 175 μs, RepRate of 200 Hz and power of 1.2 W. In both cases, pore array size was 14 mm and pore density was 15%.

To induce HDM cutaneous sensitization, 10 μl solution containing sterile PBS containing 2 μg of HDM (Greer, Item: XPB82D3A2.5, Lot No.151776) and/or 1 μg of recombinant mouse IL-1β (Biolegend, Cat No. 575106), or 1 μg of recombinant mouse TSLP (R&D System, 555-TS-010/CF) were applied on ears immediately following LMP of 30 μm (for e.c. sensitization) or of 91 μm (for d.c. sensitization) at the time points indicated in the experimental schemes in the Figures. Non-sensitized or PBS-treated mice were used as controls. All the mice were then challenged intranasally (i.n.) with HDM (2 μg) for 4 consecutive days.

## Antibody administration
To deplete Gr-1$^{hi}$ and/or Gr-1$^{int}$ cells, wildtype Balb/c mice were intraperitoneally (i.p.) injected with 100 μg of NIMP-R14 [clone NIMP-R14, depleting Gr-1$^{hi}$ (Ly-6G$^+$) and Gr-1$^{int}$ (Ly-6C$^+$/Ly-6G$^-$) cells] or 200 μg of anti-Ly6G antibody [clone 1A8, BioXCell, depleting Gr-1$^{hi}$ (Ly-6G$^+$) cells] at the time points shown in Figures. As control, mice were injected with PBS.

## Bronchoalveolar lavage (BAL) cell analyses
BAL was taken in anaesthetized mice, by instilling and withdrawing 0.5 ml of saline solution (0.9% NaCl, 2.6 mM EDTA) in the trachea. After six times lavages, BAL fluid was pooled and centrifuged, and cell number was counted using a Neubauer hemocytometer. 200 μl of BAL fluids with $2.5 \times 10^5$ cells/ml were used to prepare cytospin slides, which were then stained with Hemacolor kit (Merck, Cat No. 1116740001) to identify macrophages, lymphocytes, neutrophils and eosinophils. After counting for each cell type to obtain their frequencies, number of each cell type was calculated according to the total BAL cell numbers.

## Airway responsiveness to methacholine
Airway responsiveness to aerosolized methacholine (MCh; A2251, Sigma-Aldrich) were assessed using the forced oscillation technique (FlexiVent, SCIREQ, Montreal, Canada)[62]. Mice were anesthetized with an i.p. injection of xylasine (15 mg/kg), followed 10 min later by an i.p. injection of pentobarbital sodium (54 mg/kg). The trachea was exposed and an 18-gauge metal needle was inserted into the trachea. Airways were connected to a computer-controlled small animal ventilator, and quasi-sinusoidally ventilated with a tidal volume of 10 mL/kg at a frequency of 150 breaths/min and a positive end expiratory pressure of 2 cm H$_2$O to achieve a mean respiratory volume close to that of spontaneous breathing. For baseline measurement, each mouse was challenged for 10 s with an aerosol of PBS generated with an in-line nebulizer and administered directly through the ventilator. Then, aerosolized MCh at 50 mg/mL was administered for 10 s. The effect of MCh was calculated as the peak response, i.e., the mean of the three maximal values integrated for the calculation of lung resistance (R$_L$, cm H$_2$O s mL$^{-1}$).

## Quantitative RT-PCR
Total RNA was extracted from BAL cells using NucleoSpin RNA XS kit (Macherey-Nagel, Cat No. 740902.50) according to the manufacturer's instructions. RNA was reverse transcribed by using random oligonucleotide hexamers and amplified by means of quantitative PCR with a LightCycler 480 (Roche Diagnostics, Indianapolis, Ind) and the Light-Cycler 480 SYBR Green kit (Roche, Cat No. 04707516001), according to the manufacturer's instructions. Relative RNA levels were calculated with hypoxanthine phosphoribosyl-transferase (HPRT) as an internal control. For analyses of each set of gene expression, an arbitrary unit of 1 was given to the samples with the highest level, and the remaining

samples were plotted relative to this value. Sequences of PCR primers are: HPRT (TGGATACAGGCCAGACTTTG; GATTCAACTTGCGCTCATCTTA, 161 bp); IL-4 (GGCATTTTGAACGAGGTCAC; AAAATATGCGAAGCACCTTGG, 132 bp); IL-5 (AGCACAGTGGTGAAAGAGACCTT; TCCAATGCATAGCTGGTGATTT, 117 bp); IL-13 (GGAGCTGAGCAACATCACACA; GGTCCTGTAGATGGCATTGCA, 142 bp), MCPT8 (GTGGGAAATCCCAGTGAGAA; TCCGAATCCAAGGCATAAAG, 160 bp); CCR3 (TAAAGGACTTAGCAAAATTCACCA; TGACCCCAGCTCTTTGATTC, 150 bp).

## Serum immunoglobulin determination
For HDM-specific immunoglobulins, microtiter plates were coated with HDM and then blocked with BSA. Serum samples were incubated in the coated plates overnight at 4 °C followed by incubation with a biotinylated rat anti-mouse IgE (1:250, BD Biosciences; Cat No. 553419; clone R35-118) or IgG1 (1:250, BD Biosciences; Cat No. 553441; clone A85-1). Extravidin horseradish peroxidase (1:1000, Sigma, Cat No. E2886) and TMB (tetramethylbenzidine) Substrate Reagent Set (BD Biosciences, Cat No. 555214) were used for detection. Serum levels of HDM-specific IgG1 and HDM-specific IgE were calculated relevant to a pre-prepared serum pool from HDM-sensitized and challenged mice and expressed as arbitrary units.

## TSLP and IL-1β protein level determination
Mouse skin was chopped and homogenized with a Mixer Mill MM301 (Retsch, Dusseldorf, Germany) in lysis buffer (25 mmol/L Tris pH 7.8, 2 mmol/L EDTA, 1 mmol/L dithiothreitol, 10% glycerol, and 1% Triton X-100) supplemented with protease inhibitor cocktail (Roche, Cat No. 11873580001). Protein concentrations of skin extract were quantified by using the Bio-Rad Protein Assay (Bio-Rad Laboratories, Hercules, Calif, Cat No. 500-0006). TSLP and IL-1β levels in skin extracts were determined using the DuoSet ELISA Development Kits (R&D Systems, Minneapolis, Minn, Cat No. DY555 for TSLP, Cat No. DY401 for IL-1β).

## Cell preparation for flow cytometry analyses
For the preparation of dermal cells, ears were split into ventral and dorsal halves and incubated 1 h at 37 °C with 4 mg/ml Dispase (Gibco). Dermis was separated from epidermis and incubated 1 h at 37 °C with 1 mg/ml collagenase D (Roche), 0.25 mg/ml DNase I (Sigma) and 2.5% of foetal calf serum in PBS. Cells were passed through a 70 μm strainer (Falcon) and resuspended in FACS buffer (1% of FCS + 2 mM EDTA in PBS) and used for FACS staining.

For cell preparation of whole skin, ears were cut and incubated 1h30 at 37 °C with 0.25 mg/ml Liberase TL (Roche), 0.5 mg/ml DNase I in RPMI basic medium. Cells were passed through a 70 μm strainer, resuspended in FACS buffer and used for staining.

For cell preparation of EDLNs, EDLNs were dissociated with piston, passed through a 70 μm strainer and resuspended in FACS buffer, counted and used for FACS staining.

## Flow cytometry analyses
For surface staining, $2 \times 10^6$ skin cells or LN cells were first incubated with anti-CD16/CD32 antibody [0.5:25 (volume of antibody: volume of staining FACS buffer); clone 93, eBioscience] to block unspecific binding, followed by surface staining with the following fluorochrome-conjugated antibodies in FACS buffer: CD45 APC-eFluor780 (0.06:25, clone 30-F11), CD45R/B220 APC (1.2:25, clone RA3-6B2), GL7 PE (1.25:25, clone GL-7), Gr-1 PE (Ly-6G/Ly-6C) (0.02:25, clone RB6-8C5), CD8a PerCP-Cy5.5 (0.5:25, clone 53-6.7), TCRβ PerCP-Cy5.5 (1:25, clone H57-597), CD3 FITC (1:25, clone 145-2C11), Ly-6C PE-Cy7 (0.3:25, clone HK1.4), Ly-6G APC (1:25, clone 1A8-Ly6g), CD49b biotin (0.5:25, clone DX5) and streptavidin APC (0.5:25) were from eBioscience; Gr-1 FITC (0.05:25, Ly-6G/Ly-6C) (clone RB6-8C5), Siglec-F PE (0.5:25, clone E50-2440), CD95 PE-Cy7 (1:25, clone Jo2), CD19 FITC (1:25, clone 1D3), CXCR5 biotin (1.5:25, clone 2G8), IgE biotin (0.5:25, clone R35-72) and streptavidin BV605 (0.5:25) were from BD Biosciences; TCRβ PE-Cy7

(0.5:25, clone H57-597), CD4 BV421 (0.5:25, clone GK1.5), PD-1 PE-Cy7 (2:25, clone RMP1-30), CD45R/B220 PE-Cy7 (1.2:25, clone RA3-6B2), IgG1 PerCP-Cy5.5 (1:25, clone RMG1-1) were from Biolegend.

For IL-1β intracellular staining, cells were first stained for surface markers and then stained for IL-1β using Fixation/Permeabilization Kit (BD Biosciences, Cat No. 554715). Briefly, cells were fixed and permeabilized with Fixation/Permeabilization solution for 20 min. After wash and centrifugation, cells were resuspended in Perm/Wash buffer containing anti-IL-1β PE antibody (2:100, clone 166931, R&D Systems) for 30 min. Cells were washed and resuspended in FACS buffer for analyses.

To eliminate dead cells, propidium iodide was used for surface staining, and Fixable Viability Dye eFluor 506 (0.1:100, eBioscience Cat. 65-0866-18) was used for intracellular staining. Samples were passed on LSRFortessa X-20 (BD) and data were collected with BD FACS DIVA v8 and analysed with FlowJo.

## Histopathology
Mouse ears and lungs were fixed in 4% paraformaldehyde overnight at 4 °C and embedded in paraffin. 5 μm sections were stained with hematoxylin & eosin (H&E). For periodic Acid Schiff (PAS) staining, slides were incubated with 0.5% aqueous periodic acid (Alfa Aesar), washed with water and incubated 15 min in Schiff's reagent (Merck). Slides were counterstained with hematoxylin and differentiated with acid alcohol.

## Immunohistochemistry
For immunohistochemistry (IHC) staining of major basic protein (MBP) and mast cell protease 8 (MCPT8), 5μm paraffin sections were treated with 0.6% $H_2O_2$ to block endogenous peroxidase activity before antigen retrieval with either Pepsin (Life technologies; for IHC of MBP) or citric buffer (10 mmol/L citric acid, pH 6; for IHC of MCPT8). Slides were then blocked with normal rabbit serum (Vector Laboratories) and incubated overnight with rat anti-mouse MBP (1:2000, provided by Dr James J Lee, Mayo Clinic, Rochester) and rat anti-mouse MCPT8 (1:500, clone TUG8, Biolegend). Slides were then incubated with biotinylated rabbit anti-rat IgG (1:300) and treated with AB complex (Vector Laboratories, Cat No. PK-6104). Staining was finally visualized with AEC high-sensitivity substrate chromogen solution (Dako) and counter-stained with hematoxylin. Slides were scanned with Nanozoomer 2.0 HT (Hamamatsu) using the program NDP.scan, and images were viewed with NDP.view 2.

## RNAscope in situ hybridization
To localize TSLP and IL-1β RNA in the skin, in situ hybridization was performed on freshly prepared 5μm paraffin sections with the RNAscope 2.5 FFPE Red detection Kit (Advanced Cell Diagnostics, Hayward, CA, USA, Cat No. 322360), according to the manufacturer's protocol. Mm-Ppib probe (Mus musculus peptidylprolyl isomerase B; Cat No. 313917) was used as a positive control, and DapB probe (Bacterial Bacillus subtilis dihydrodipicolinate reductase; Cat No. 310043) was used as a negative control. Probe-Mm-TSLP (Cat No. 432741) and Probe- Mm-Il1b (Cat No. 316891) were used for detection of TSLP and IL-1β, respectively.

## Statistical analyses
Data were analysed using GraphPad Prism 9. Comparison of two groups was performed either by Student's two-tailed unpaired t-test with Welch's correction or the two-tailed Mann–Whitney rank sum nonparametric test depending on results from the Kolmogorov–Smirnov test for normality. Comparison of more than two samples was performed by ordinary one-way ANOVA followed by Tukey's post hoc test. Data show values from individual mice and are presented with mean ± SEM (for Student's t-test or one-way ANOVA), or with median (for Mann–Whitney rank sum nonparametric test).

The p values are marked in the Figures. $p > 0.05$ is considered as non significant.

## Reporting summary
Further information on research design is available in the Nature Research Reporting Summary linked to this article.

## Data availability
All other data are available in the article and its Supplementary files or from the corresponding author upon reasonable request. Source data are provided with this paper.

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

## Acknowledgements

We thank the staff of animal facilities, mouse supporting services, flow cytometry, histopathology, microscopy and imaging, and cell culture of IGBMC and Institut Clinique de la Souris (ICS) for excellent technical assistance. We are grateful for W. Paul for providing 4C13R dual reporter mice. We thank J. Heller and J. Demenez for helping with genotyping and histology analyses. We would like to acknowledge the funding supports from l'Agence Nationale de la Recherche (ANR-19-CE17-0017; ANR-19-CE17-0021) to M.L., from Fondation Recherche Medicale (Equipes FRM 2018) to M.L., and the first joint programme of the Freiburg Institute for Advanced Studies (FRIAS) and the University of Strasbourg Institute for Advanced Study (USIAS) to M.L. This work of the Interdisciplinary Thematic Institute IMCBio, as part of the ITI 2021-2028 program of the University of Strasbourg, the Centre National de la Recherche Scientifique (CNRS) and the Institut National de la Santé et de la Recherche Médicale (Inserm), was supported by IdEx Unistra (ANR-10-IDEX-0002), and by SFRI-STRAT'US project (ANR 20-SFRI-0012) and EUR IMCBio (ANR-17-EURE-0023) under the framework of the French Investments for the Future Program. J.S., Y.W., P.M. and B.G. were supported by PhD fellowships from Equipes FRM 2018, the Association pour la Recherche à l'IGBMC (ARI), Region Alsace, and International PhD Program from LabEx INRT funds.

## Author contributions

J.S., W.Y., and M.L. conceived and designed the study. J.S. and W.Y. conducted most experiments and acquired data. P. Marschall contributed to the establishment of e.c and d.c sensitization models and the set up for flow cytometry analyses; F.D. and C.L. performed airway function study; B.G. contributed to IL-1 detection by RNAscope and ELISA analyses; B.G., P. Meyer, P.H., C.H. and E.F. contributed to the analyses for BAL cells and immune cells by flow cytometry; M.G. and L.C. contributed to the experiments during the revision; S.M. contributed NIMP-R14 clone; M.O.-A. prepared NIMP-R14 Ab for in vivo administration. J.S., W.Y. and M.L. analysed and interpreted data. J.S., W.Y. and M.L. wrote and revised the manuscript. M.L. directed the study and supervised the work.

## Competing interests

The authors declare no competing interests.
