## [Peer Review File · Nature Communications]

Context-dependent role of TSLP and IL-1 β in skin allergic sensitization and atopic marchREVIEWER COMMENTS

Reviewer #1 (Remarks to the Author):

In this study, Segaud et al investigated the role of TSLP in epicutaneous sensitization (e.c.) or dermacutanueous sensitization (d.c.) and consequently the development of allergic asthma (atopic march) by using an innovative laser-assistant microporation (LMP) system. Importantly, they identified increased levels of IL-1 β induced by a deeper disruption of the skin that is a major driver for skin allergic sensitization and atopic march, which is independent of TSLP. In general, the study is interesting and the manuscript is well written. However, given that some of the findings have been previously reported by the same group, including the role of TSLP in skin allergic sensitization and atopic march [e.g., Leyva-Castillo JM et al. *J Invest Dermatol* 133, 154-163 (2013)] and TSLP-promoted TFH/TH2 differentiation in AD pathogenesis and subsequent asthmatic phenotype with the LMP system [Marschall, et al. *J Allergy Clin Immunol* 147, 1778-1794 (2021)]. Thus, the novelty of this study is certainly a concern. Although different allergens were used, the rationale, experimental approaches, and major findings are almost similar. The authors specifically investigated IL-1 β that is induced by a deeper disruption of the skin, but no evidence was shown to support a direct link between TSLP and IL-1 β for the skin allergic sensitization and the atopic march. In fact, they appear to be two independent studies (TSLP, IL-1 β). Specifically:

1. In the previous work, the authors have generated mouse models with skin barrier disruption at different anatomic depths; 11 μ m and 30 μ m-LMP, for skin allergic sensitization and the atopic march. In the present study, both 30 μ m and 91 μ m-LMP were used to represent skin barrier disruption in epidermis and demis by using different experimental settings. However, it is not clear how the depth was determined, how the heterogeneity was controlled, and whether the experimental models truly mimic the cutaneous allergen sensitization. Furthermore, what is the rationale for the mouse model with skin barrier disruption in the ear skin rather than back skin?

2. TSLP is increased in LMP-30 μ m and 91 μ m, but there is no difference between LMP-30 and LMP-91 μ m. However, with the increased expression of TSLP (300-450 pg/mg), there was no eosinophil and basophils detected in skin tissues (Fig 1, E), suggesting that TSLP derived from epidermis and demis may not be the major driver for the increased eosinophils and basophils observed in HDM-treated mouse model. In Fig 1E, the reduced eosinophils and basophils observed in *Tslp*^{-/-} mice may be due to the deficiency of TSLP in other immune cells, e.g., DCs, ILC2, of the skin tissues. To show the role of the epidermis or dermis-derived TSLP in driving downstream responses, the TSLP conditional knockout mice rather than global TSLP knockout mice (*Tslp*^{-/-}) may help.

3. The authors nicely showed that epicutaneous HDM sensitization induced a TSLP-dependent TFH/TH2 differentiation and GC response. TFH is critical in determining the type of antibody produced. Thus, the reduced *Tfh* observed in *Tslp*^{-/-} mice may lead to antibody defects. However, no changes were observed in this study for the downstream IgG1⁺ and IgE⁺ B cells (Fig 1, I) and subsequent production of HDM-specific IgG1 and IgE (Fig 1, J), which are the major phenotypic changes for AD. In other words, no correlation was noted for TFH and types of B cells and antibodies.

4. For the HDM-induced asthmatic inflammation, both eosinophils and macrophages are the predominant cell types in BALs (Fig 2B). However, the authors detected Th2 cytokines in those BAL cells by RT-PCR. It is unclear why the Th2 cytokines were not tested in BAL fluids by using ELISA, a routine approach to assess the Th2-associated lung inflammation. Also, it is not clear the reason why the basophils were specifically investigated. Any roles of basophils in TSLP-driven skin allergic inflammation and atopic march?

5. IL-1 β was specifically detected with interest in the skin of the AD-like mouse model. While an increased IL-1 β was noted following a deeper disruption of the skin and IL-1 β does exacerbate HDM-induced skin sensitization, which may be the one but not the major one contributing to the downstream skin sensitization and the atopic march. In addition to mouse ear skin, does the mouse

back skin show a similar pattern for the expression of TSLP and IL-1 β following skin different depths of disruption?

6. It is no surprise to see that IL-1 β promotes allergic sensitization and atopic march independent of TSLP. However, it would be of interest to see the possible cross-talk between TSLP and IL-1 β . In fact, a recent study has suggested that IL-1 β may be an early key mediator for the acquisition of an AD phenotype through induction of TSLP and alteration of the epidermal homeostasis (Bernard et al. *J Pathol* 2017; 242(2); 234-245).

7. To generate a stable deletion of neutrophils and monocytes/macrophages, repetitive injections are needed for long-term experiments with the right doses for NIMP-R14 and anti-Ly6G. Particularly, Neutrophil counts have been shown to return to basal levels about 3 days post NIMP-R14 injection (Stachowicz et al. *Front in Immunol* 2020). However, in this study, only two doses (D1 and D2) were given to the mouse model and both neutrophils and macrophages were noted in BALs of NIMP-R14 or anti-Ly6G pre-treated mice (Fig 6, B). Thus, the authors should provide evidence to show the consistent deletions under the current experimental protocol (Fig 6, A and Fig 7, A).

8. The data on IL-1 β blockade with anti-IL-1 β or Anakinra in supplementary figure S7 is very interesting, which provides direct evidence for the significance of IL-1 β in allergic sensitization and the atopic march. Thus, instead of showing the data on the deletion of cellular sources of IL-1 β (Fig 6 and 7), it is better to have a comprehensive study on the direct blockage of IL-1 β . It is clear that IL-1 β is not the only inflammatory mediator secreted by neutrophils and monocytes/macrophages.

9. In all experimental protocols, the dose used for experiments should be clearly illustrated, including allergen, antibodies.

10. Overall sample size is relatively small. e.g., n=2, n \geq 3, n \geq 4, plus the spread in data, suggesting a big variation among these individuals. Thus, the statistical power may not be sufficient to detect any but the largest differences.

Reviewer #2 (Remarks to the Author):

Segaud et al. investigated the role of the epithelial cytokine thymic stromal lymphopoietin (TSLP) in shaping the pathology of allergic skin inflammation by establishing unique experimental systems that mimic atopic dermatitis using Laser Micro Poration devices.

First, the authors demonstrated the requirement of epidermis TSLP in the induction of HDM-induced skin inflammation. They next showed that deeper skin barrier disruption resulted in IL-1 β production by neutrophils, and macrophages infiltrated in the dermis; furthermore, IL-1 β sensitized T and B cells, resulting in the exacerbation of type 2 airway inflammation. Thus, IL-1 β is a new potential therapeutic target in severe atopic dermatitis.

The manuscript is well written, and the logical flow is clear, but many results need additional experiments to support their conclusions. In particular, the authors used experimental systems with two-week protocols, but the "atopic march" is a concept involving the course of allergic disease over a longer period of time. Thus, it is difficult to draw a connection between the results presented in this manuscript and the pathogenesis of "atopic march".

Major Comments:

1. In Figure. 1F to 1J, 2A-C, S2A, 3E-F, 7E, the sample size should be increased. For example, the sample size in the WT d.c. PBS group was only 1 to 2. Furthermore, no significant differences were found in some of the experimental results.

2. In Figure 1E, 2D, 3D, 4D, and 6D, the authors should evaluate their results using a (semi-)quantitative analysis.

3. In Figures 3 and 4, the authors claimed that TSLP and IL-1 β act independently based on experiments that showed the IL-1 β -enhanced immune response in TSLP-deficient mice. However, they did not mention the possibility that TSLP affected IL-1 β production. For example, the IL-1 β production in TSLP-deficient mice should be assessed when they are subjected to laser-assisted microporation (LMP).

4. In Figure 5A, the authors showed that GR-1hi neutrophils were induced subcutaneously when d.c. was performed, unlike e.c. They should therefore compare e.c. and d.c. to determine the factors that induce subcutaneous infiltration of neutrophils in vivo.

5. For a better understanding of the IL-1 β function, it is better to clarify whether IL-1 β acts locally or systemically and how long its expression remains elevated in the skin or body. It will be useful to investigate the dynamics of IL-1 β levels in the blood over time after d.c. sensitization.

Minor Comments:

1. The term "GR-1" in the manuscript should be rendered "Gr-1".

2. In Figure 4A, "IL-1 β " seems to be incorrectly described as "IL1 β ".

3. In Figure 5A, the black line in the FACS results for "Non-treated" seems to be incorrectly included.

Reviewer #3 (Remarks to the Author):

Major comments

This is a carefully conducted work on the impact of the mode of cutaneous sensitization on the Th2 response to antigen and the subsequent atopic march that leads to allergic lung inflammation.

The experiments are rigorously conducted and the conclusions are strongly backed up by the data presented.

This manuscript advances the field of allergic diseases in general and atopic dermatitis in particular

The manuscript could benefit by addressing the comments below.

Specific comments

1. It would be desirable but not imperative to examine airway hyperresponsiveness in mice differentially exposed to cutaneous sensitization by the e.c. versus d.c. route

2. The authors report a decrease in IgG1 antibody levels in e.c. sensitized TSLPKO mice compared to WT controls. However, the difference is not statistically significant.

3. The authors report a partial effect of Tslp deletion on the response to d.c. sensitization. Could this simply be due to the fact that this mode of sensitization results in a more rigorous response. Also, this "partial effect" is equivalent to the effect on e.c. sensitization in some assays such as the percentage of Tfh cells and IL-4+Tfh cells.

4. The authors have shown that administration of IL-1b does not induces TSLP expression. It would be important to show whether TSLP could induce IL-1b expression

5. The authors have shown that e.c. HDM sensitization and to a much larger extent d.c. HDM sensitization induces the infiltration of Il1b expressing GR-1hi and GR-1int cells in the skin. Is this induction dependent of TSLP? This would help strengthen the conclusion that IL-1b expressing neutrophils and monocytes/macrophages promotes skin inflammation in a TSLP-independent manner.

6. Mice d.c sensitized with HDM show stronger airway inflammation compared to mice e.c. sensitized with HDM, evidenced by a greater Th2 and eosinophilic response. There were also more neutrophils in the airways of d.c. sensitized mice compared to e.c. sensitized mice. Given the role of neutrophils in severe asthma is the severe airway inflammation in d.c. sensitized mice associated with AHR and is it corticosteroid-sensitive?

7. Neutrophils are essential for host defense against microbial pathogens by producing antimicrobial molecules. AD patients are often colonized with S. aureus. Blocking of Il1b or depleting GR1+ hi or GR1+int cells could be detrimental as it may impair S aureus clearance. The authors need to include this potential limitation in the Discussion of the potential usefulness of IL-1 blockade and granulocyte depletion strategies in the treatment of asthma in AD patients.

Point_to_Point_Response:

REVIEWER COMMENTS

Reviewer #1 (Remarks to the Author):

In this study, Segaud et al investigated the role of TSLP in epicutaneous sensitization (e.c.) or dermacutanueous sensitization (d.c.) and consequently the development of allergic asthma (atopic march) by using an innovative laser-assistant microporation (LMP) system. Importantly, they identified increased levels of IL-1 β induced by a deeper disruption of the skin that is a major driver for skin allergic sensitization and atopic march, which is independent of TSLP. In general, the study is interesting and the manuscript is well written. However, given that some of the findings have been previously reported by the same group, including the role of TSLP in skin allergic sensitization and atopic march [e.g., Leyva-Castillo JM et al. J Invest Dermatol 133, 154-163 (2013)] and TSLP-promoted TFH/TH2 differentiation in AD pathogenesis and subsequent asthmatic phenotype with the LMP system [Marschall, et al. J Allergy Clin Immunol 147, 1778-1794 (2021)]. Thus, the novelty of this study is certainly a concern. Although different allergens were used, the rationale, experimental approaches, and major findings are almost similar. The authors specifically investigated IL-1 β that is induced by a deeper disruption of the skin, but no evidence was shown to support a direct link between TSLP and IL-1 β for the skin allergic sensitization and the atopic march. In fact, they appear to be two independent studies (TSLP, IL-1 β). Specifically:

Answer: We thank the reviewer to recognize the importance of our study for understanding the allergic sensitization and atopic march using innovative experimental system.

Concerning the novelty of the current study:

We [Leyva-Castillo JM et al. J Invest Dermatol 133, 154-163 (2013)] and others have previously reported that TSLP is induced by skin barrier disruption in mouse or human and promotes ovalbumin (OVA)-induced Th2-type sensitization through the tape-stripped skin and the subsequent asthma in mice. More recently, we have provided evidence that skin TSLP plays an important role in promoting epicutaneous OVA-induced Tfh cells, which provide critical B cell help in the GC for the generation of allergen-specific IgE [Marschall, et al. J Allergy Clin Immunol 147, 1778-1794 (2021)]. However, despite of these pieces of evidence suggesting that TSLP could be an important target for AD therapy and for preventing the atopic

march, it had not yet been explored the role of TSLP in allergen sensitization occurring in AD skin with different severity, reflected by varied skin barrier defects due to various genetic or/and environmental causes, or at different stages of the diseases.

Taking advantage of the powerful LMP system that we established to disrupt the targeted skin layers at a precise anatomic depth of mouse skin, we were able to model allergen exposure superficially (30 μ m) or deeply (91 μ m) in the skin, leading to e.c. sensitization or d.c. sensitization in the current study. We delineated the context-dependent role of TSLP in skin allergic sensitization and atopic march: once skin disruption goes deeper, the requirement of TSLP is diminishes, whereas IL-1 β derived from neutrophils and monocytes takes the role. We think the methodology and the findings highlight the novelty of this paper, which should be of importance not only for understanding how allergen sensitization is regulated by skin microenvironment and cytokine signaling, but also for the current development of biologic medicine for atopic diseases. We believe that our findings will provide important insight for the recent clinical trial results in AD obtained from TSLP neutralizing antibody Tezepelumab, for example, moderate to severe AD adults treated with Tezepelumab presented only a numeric but not significant improvement compared to placebo-treated group.

Concerning “No evidence is supported a direct link between TSLP and IL-1 β ”: Yes, our data show that TSLP and IL-1 β are indeed two independent factors. Please see below the detailed response to the point 6.

1. In the previous work, the authors have generated mouse models with skin barrier disruption at different anatomic depths; 11 μ m and 30 μ m-LMP, for skin allergic sensitization and the atopic march. In the present study, both 30 μ m and 91 μ m-LMP were used to represent skin barrier disruption in epidermis and dermis by using different experimental settings. However, it is not clear how the depth was determined, how the heterogeneity was controlled, and whether the experimental models truly mimic the cutaneous allergen sensitization. Furthermore, what is the rationale for the mouse model with skin barrier disruption in the ear skin rather than back skin?

Answer: According to the manufacturer of P.L.E.A.S.E. machine, ablation depth in (μ m) is explained as a function of fluence (J/cm²). Ablation depth increases linearly with the fluence (PMID: 24500855). The factor of around 4 of the fluence is used by the manufacturer to determine the depth with this device. Depth penetration into tissue (μ m) are (ideal) theoretical values; their corresponding ablation effects on mouse skin are evaluated and controlled by histological analyses of the skin upon the laser microporation, as shown in previous publications from us (PMID: 33068561) and others (PMID: 25941327).

The fluence is calibrated in Pantec machine maintenance process every year and the specified tolerance is \pm 15%, so also the depth precision is \pm 15%.

We evaluated LMP on mouse ear and back skin. Our choice to use ear skin rather than dorsal skin to model e.c. and d.c. sensitization was mainly based on the reproducibility of skin LMP: there is no need to remove hair for ear LMP, but the LMP on back skin requires the removal of hair. Not only that hair removal by depilation cream brings additional factors to impact skin physiology, but also the hair cycle on the back skin adds the variation of the allergen sensitization. Nevertheless, we provide here **Fig1_To_Reviewer1**, showing an example that similar observations were obtained with d.c. sensitization on microporated dorsal skin as ear skin.

2. TSLP is increased in LMP-30 μ m and 91 μ m, but there is no difference between LMP-30 and LMP-91 μ m. However, with the increased expression of TSLP (300-450 pg/mg), there was no eosinophil and basophils detected in skin tissues (Fig 1, E), suggesting that TSLP derived from epidermis and dermis may not be the major driver for the increased eosinophils

and basophils observed in HDM-treated mouse model. In Fig 1E, the reduced eosinophils and basophils observed in *Tslp*^{-/-} mice may be due to the deficiency of TSLP in other immune cells, e.g., DCs, ILC2, of the skin tissues. To show the role of the epidermis or dermis-derived TSLP in driving downstream responses, the TSLP conditional knockout mice rather than global TSLP knockout mice (*Tslp*^{-/-}) may help.

Answer: First, yes, TSLP induction by 30 μm LMP in the skin at this level is not sufficient on its own (without allergen, e.c. PBS, **Fig. 1e**) to induce a clear infiltration of eosinophil and basophil in skin; however, once there is allergen (HDM), this (e.c. HDM, **Fig. 1e**) induces a clear infiltration of eosinophils and basophils, which crucially requires TSLP, because such infiltration is abolished in *Tslp*^{-/-} mice. *In other words*, TSLP is not sufficient but necessary for e.c. HDM-induced eosinophil and basophils. It suggests that TSLP acts as a crucial “adjuvant” for allergen sensitization, possibly by acting on dendritic cell-T cell differentiation axes (PMID: 33068561).

Second, we showed that upon LMP_30μm and LMP_91μm, TSLP was only detected in epidermis by RNAscope in situ hybridization (an approach with high sensitivity and specificity), and no signal for TSLP was detected in the dermis (**Fig. 1c**). 1) This is in agreement with our previous report (PMID: 22832486) showing that keratinocyte-derived TSLP is induced upon barrier disruption by tape-stripping. Indeed, LMP-30μm induced a similar level of TSLP (200-500 pg/total protein (mg)) as tape-stripping. In our previous report (PMID: 22832486), we showed that TSLP induction by tape stripping was abolished in *Tslp*^{ep-/-} (K14-Cre/*Tslp*^{L2/L2} mice), and tape-stripping-OVA-induced allergen sensitization was diminished in *Tslp*^{ep-/-} mice. 2) Because of difficulties and limitations due to the pandemic, particularly on mouse breeding for the maintenance of certain mouse lines and the generation of cohort mice (e.g. K14-Cre/*Tslp*^{L2/L2} line), we could not re-perform the LMP HDM sensitization on such mice. However, even though it cannot be excluded that TSLP may in any case be expressed by other cells at a much smaller level compared to epidermis, we think that *Tslp*^{-/-} mice are better justified for the current study. This is because our aim was to investigate context-dependent role of TSLP in allergen sensitization occurring at different barrier disruption depth, which we hope to bring insights for why TSLP targeting (e.g. current neutralization antibody blockade in clinical trial) is not sufficient for severe AD and atopic march, and what other factors could be identified as new targets.

3. The authors nicely showed that epicutaneous HDM sensitization induced a TSLP-dependent TFH/TH2 differentiation and GC response. TFH is critical in determining the type of antibody produced. Thus, the reduced Tfh observed in *Tslp*^{-/-} mice may lead to antibody defects. However, no changes were observed in this study for the downstream IgG1+ and IgE+ B cells (Fig 1, I) and subsequent production of HDM-specific IgG1 and IgE (Fig 1, J), which are the major phenotypic changes for AD. In other words, no correlation was noted for TFH and types of B cells and antibodies.

Answer: We guess that there was a misunderstanding for our results from the two contexts. We showed: 1) In e.c. HDM sensitization, the induced Tfh/Th2 differentiation and GC response are abolished in *Tslp*^{-/-}, which are accompanied by a highly reduced GC B cells, IgE+ B cells, IgG1+ B cells, as well as a reduced HDM-specific IgE (significantly) and HDM-specific IgG1 (tendency). These phenotypic changes are correlated together, as expected. 2) In d.c. HDM sensitization, the induced Tfh/Th2 differentiation and GC response are not anymore abolished in *Tslp*^{-/-}, despite certain reduction. IgE+ B cells, IgG1+ B cells, as well as serum HDM-specific IgE and HDM-specific IgG1 are comparable between *Tslp*^{-/-} and WT, indicating that TSLP is less required for these phenotypes in the context of d.c. sensitization compared to e.c. sensitization.

We guess that the reviewer asked in d.c. HDM sensitization context, why Tfh/GC number has certain reduction in *Tslp*^{-/-} mice, but serum HDM-IgE and HDM-IgG1 are not. We suspect that as the induction of Tfh/GC cells is partially reduced but not totally abolished, this may explain the comparable serum IgE and IgG1 levels between *Tslp*^{-/-} and WT mice, as the rest levels of Tfh/GC in *Tslp*^{-/-} mice could be sufficient for the observed HDM-IgE/G1.

4. For the HDM-induced asthmatic inflammation, both eosinophils and macrophages are the predominant cell types in BALs (Fig 2B). However, the authors detected Th2 cytokines in those BAL cells by RT-PCR. It is unclear why the Th2 cytokines were not tested in BAL fluids by using ELISA, a routine approach to assess the Th2-associated lung inflammation. Also, it is not clear the reason why the basophils were specifically investigated. Any roles of basophils in TSLP-driven skin allergic inflammation and atopic march?

Answer: Although the detection of Th2 cytokines in BAL fluids by ELISA is a routine approach, the sensitivity is low. The R&D Elisa kits that we used has the detection limit for IL-4 at 7.8-15.6 pg/ml and for IL-13 at 15.6 pg/ml. A number of studies showed that Th2 cytokine levels in BAL fluids with different sensitization models (for example PMID: 22355542; PMID: 31823765; PMCID: PMC7649385) are close or under such detection limits. This was also the case for our model. We therefore employed quantitative RTPCR to analyze BAL cells, which has a better sensitivity for detecting and comparing the levels of Th2 cytokines.

Basophils are an important cellular component in type 2 immunity. The role of basophils has been reported not only in the skin but also in airways. These cells were shown to infiltrate the lung of asthma patients (PMID: 11496235; PMID: 10629459; PMID: 34321876). In OVA-induced allergic mouse model, it has been reported that basophils are recruited in the BAL and promote airway hyperresponsiveness (PMID: 29777594). Concerning the relationship between skin TSLP and basophils, it has been shown in one report that TSLP-mediated epicutaneous inflammation promotes allergic diarrhea and anaphylaxis, and depletion of basophils reduces development of gastrointestinal allergy (PMID: 25365222), and in another report that skin TSLP promotes eosinophilic esophagitis (EoE), and basophils depletion ameliorates established EoE-like disease (PMID: 23872715). These studies thus suggested a role for basophils in skin TSLP-promoted progression from AD to food allergies.

Therefore, although basophils are not routinely examined by differential counting of BAL cells, these cells deserve to be included in asthmatic phenotype analyses. MCPT8 RNA level in BAL cells, as we showed in the manuscript, is a sensitive method to examine basophils in BAL.

5. IL-1 β was specifically detected with interest in the skin of the AD-like mouse model. While an increased IL-1 β was noted following a deeper disruption of the skin and IL-1 β does exacerbate HDM-induced skin sensitization, which may be the one but not the major one contributing to the downstream skin sensitization and the atopic march. In addition to mouse ear skin, does the mouse back skin show a similar pattern for the expression of TSLP and IL-1 β following skin different depths of disruption?

Answer: We conclude that IL-1 β is an important factor implicating in d.c. HDM sensitization and the subsequent atopic march based on several pieces of evidence: 1) IL-1 β is highly increased in d.c. HDM compared to e.c. HDM-treated skin; 2) depletion of IL-1 β -expressing Gr-1 cells during the d.c. HDM sensitization phase reduces the subsequent asthma, which can be restored by administration of IL-1 β in the skin (**Fig. 6-7**); 3) supplement of IL-1 β exacerbates e.c. HDM sensitization. Thus, IL-1 β represents an important factor for allergic sensitization when it occurs at a deeper depth, but of course, IL-1 β may not be the only one contributing to skin sensitization.

Concerning mouse back skin, please refer to the answer to question-1 with **Fig1_To_Reviewer**.

6. It is no surprise to see that IL-1 β promotes allergic sensitization and atopic march independent of TSLP. However, it would be of interest to see the possible cross-talk between TSLP and IL-1 β . In fact, a recent study has suggested that IL-1 β may be an early key mediator for the acquisition of an AD phenotype through induction of TSLP and alteration of the epidermal homeostasis (Bernard et al. J Pathol 2017; 242(2); 234-245).

Answer: We thank the reviewer for raising this important point. As requested also by the other two reviewers, we have now added more data in Figures to elucidate the relationship between TSLP and IL-1 β :

1) we show that the depletion of IL-1 β -expressing neutrophils and monocytes/macrophages does not impact TSLP production (**Fig. 5g**), and the supplement of IL-1 β to mouse skin does not induce TSLP expression (**Fig. 3d**), which is indeed different from the report of Bernard, M (2017) showing that in vitro IL-1 β induces TSLP in reconstructed human epidermis culture. In addition, the supplement of IL-1 β enhances the e.c. HDM sensitization in *Tslp*^{-/-} mice similarly as in WT mice (**Fig. 3** and **Fig. 4**), indicating that the effect of IL-1 β in promoting skin allergic sensitization is TSLP-independent.

2) we show that the supplement of TSLP to mouse skin does not increase IL-1 β level (**new Fig. 5j**), and moreover, TSLP is not required for d.c. HDM-induced IL-1 β or infiltration of neutrophils and monocytes / macrophages, as IL-1 β levels and Gr-1 cells were similarly observed in *Tslp*^{-/-} and wildtype mice (**new Fig. 5h and 5i**).

Together, we conclude that IL-1 β and TSLP are two independent factors promoting allergic sensitization and atopic march. We also added a new session in the discussion dedicated to this point (**p. 18**):

7. To generate a stable deletion of neutrophils and monocytes/macrophages, repetitive injections are needed for long-term experiments with the right doses for NIMP-R14 and anti-Ly6G. Particularly, Neutrophil counts have been shown to return to basal levels about 3 days post NIMP-R14 injection (Stachowicz et al. Front in Immunol 2020). However, in this study, only two doses (D1 and D2) were given to the mouse model and both neutrophils and macrophages were noted in BALs of NIMP-R14 or anti-Ly6G pre-treated mice (Fig 6, B). Thus, the authors should provide evidence to show the consistent deletions under the current experimental protocol (Fig 6, A and Fig 7, A).

Answer: We think that the reviewer misunderstood our purpose for the experiments with NIMP-R14 or anti-Ly6G injection. We had aimed at depleting neutrophils and monocytes/macrophages only during the skin sensitization phase (but not during the airway challenge phase), in order to investigate the role of these cells in skin allergic sensitization.

As described in the result session (**p. 13**): mice were i.p. injected at D-1 and D2 with NIMP-R14 or anti-Ly6G Ab, and d.c. sensitized with HDM at D0 and D3, followed by i.n. challenge with HDM at D10-13 (**Fig. 6a**). This protocol was designed based on the previous report that Gr-1^{hi} and Gr-1^{int} cells could be efficiently depleted 1 day after the *i.p.* injection of NIMP-R14 Ab, but started to recover 4 days after (as mentioned by the reviewer). Note that we confirmed that a repeated Ab injection at D2 maintained the cell depletion during the d.c. sensitization phase, while Gr-1^{hi} and Gr-1^{int} cells were well recovered before the i.n. HDM challenge (at D9, see **Supplementary Fig. 8**), thus allowing us to conclude the role of these cells in skin sensitization phase.

8. The data on IL-1 β blockade with anti-IL-1 β or Anakinra in supplementary figure S7 is very interesting, which provides direct evidence for the significance of IL-1 β in allergic

sensitization and the atopic march. Thus, instead of showing the data on the deletion of cellular sources of IL-1 β (Fig 6 and 7), it is better to have a comprehensive study on the direct blockage of IL-1 β . It is clear that IL-1 β is not the only inflammatory mediator secreted by neutrophils and monocytes/macrophages.

Answer: Yes, we had tested whether a direct blockade of IL-1 β signalling, by administrating anti-IL-1 β Ab or Anakinra during the d.c. HDM sensitization, could reduce the subsequent asthmatic phenotype. These results were summarized in **Supplementary Fig. 9**, showing that mice with i.p. injection with anti-IL-1 β during d.c. HDM sensitization developed a relatively weaker asthmatic inflammation although HE staining did not show a striking reduction for inflammatory cell infiltration in the lung (**Supplementary Fig. 9 a-d**). On the other hand, injection of Anakinra did not reach a better reduction for lung inflammation (**Supplementary Fig. 9e-f**), suggesting that unlike the depletion of IL-1 β -expressing neutrophils and monocytes/macrophages, the blockade of IL-1 β or IL-1 signaling using the available anti-IL-1 β Ab or anakinra only reaches a mild effect in reducing the d.c. HDM sensitization-triggered allergic asthma.

We discussed these data in the discussion session (**p.17** last paragraph). We suspect that this could be due to the inefficiency of the blockade. For example, Anakinra blockade efficiency in mice seems to be influenced by genetic background, and its treatment has led to both positive and negative results in different human diseases suggesting complex role of IL-1 receptor antagonist and its usage difficulties from tissues to tissues. The ongoing efforts for developing more potent treatments than Anakinra may possibly provide better reagents for targeting IL-1 signaling.

We agree with the reviewer that IL-1 β may not be the only inflammatory mediator secreted by neutrophils and monocytes/macrophages (see discussion, **p. 20**): “even though our study has suggested that IL-1 β is one key factor mediating the promoting effect of neutrophils in allergic sensitization and the subsequent allergic asthma, it is possible that other factors derived from neutrophils including neutrophil extracellular traps may also have a contribution.”

9. In all experimental protocols, the dose used for experiments should be clearly illustrated, including allergen, antibodies.

Answer: All doses used for experiments, including allergen, antibodies are indicated in the Methods session.

10. Overall sample size is relatively small. e.g., n=2, n \geq 3, n \geq 4, plus the spread in data, suggesting a big variation among these individuals. Thus, the statistical power may not be sufficient to detect any but the largest differences.

Answer: In **Fig.1, f-i**, and **Fig. 2, a-c**, all the statistic comparisons, i.e. WT/4C13R^{Tg}_ecHDM versus Tslp^{-/-}/4C13R^{Tg}_ecHDM; WT/4C13R^{Tg}_dcHDM versus Tslp^{-/-}/4C13R^{Tg}_dcHDM; WT/4C13R^{Tg}_ecHDM versus WT/4C13R^{Tg}_dcHDM, contain data with n \geq 3 or 4. The data with n=2 were only from the control group WT/4C13R^{Tg}_dcPBS control (without allergen HDM). The reason was that during the experiment one wildtype control mouse accidentally died. As this piece of data served as PBS control, and was not used to calculate statistical significance with any of other group, we decided to keep the WT/4C13R^{Tg}_dcPBS data as it is. We did not replace WT/4C13R^{Tg}_dcPBS data with data from another experiment, as the current Fig. 1 and 2 represented one experiment containing all the different groups in the same manipulation. Indeed, it is most appropriate to present the data from one experiment with the age/sex-matched littermates, with the same experimental procedures for treatment and analyses.

Nevertheless, to show the reviewer about WT/4C13R^{Tg}_dcPBS data, we attach here one Figure (**Fig.2_To_Reviewer**) to show one independent experiment which included

WT_4C13R^{Tg}_NS; WT/4C13R^{Tg}_ecPBS; WT/4C13R^{Tg}_ecHDM; WT/4C13R^{Tg}_dcPBS; and WT/4C13R^{Tg}_dcHDM. The relationship of these 5 groups including WT/4C13R^{Tg}_dc PBS is similarly observed as **Fig. 1** and **Fig. 2** in the manuscript.

We would like to note that in the revised **Fig.1j**, **Fig. 3g** and **Fig.6e**, we removed the data from non-sensitized groups (WT_4C13R^{Tg}_NS; WT_4C13R^{Tg}_dcPBS), as in these groups without sensitization, HDM-specific IgG1 and IgE were never detected (as expected). For **Fig. 3g**, we replaced the data with another independent experiment which included more experimental mice.

Concerning relatively small numbers in mouse experiments: the composition of experimental mouse cohort is actually limited by the number of appropriate mice available, that is age and sex matched littermates which are produced from breedings. Despite such limitation for mouse number per group, all the data were reproduced with independent experiments.

Reviewer #2 (Remarks to the Author):

Segaud et al. investigated the role of the epithelial cytokine thymic stromal lymphopoietin (TSLP) in shaping the pathology of allergic skin inflammation by establishing unique experimental systems that mimic atopic dermatitis using Laser Micro Poration devices. First, the authors demonstrated the requirement of epidermis TSLP in the induction of HDM-induced skin inflammation. They next showed that deeper skin barrier disruption resulted in IL-1 β production by neutrophils, and macrophages infiltrated in the dermis; furthermore, IL-1 β sensitized T and B cells, resulting in the exacerbation of type 2 airway inflammation. Thus, IL-1 β is a new potential therapeutic target in severe atopic dermatitis. The manuscript is well written, and the logical flow is clear, but many results need additional experiments to support their conclusions. In particular, the authors used experimental systems with two-week protocols, but the “atopic march” is a concept involving the course of allergic disease over a longer period of time. Thus, it is difficult to draw a connection between the results presented in this manuscript and the pathogenesis of “atopic march”.

Answer: We thank the review for the clear resume of our discoveries of this manuscript. In the revised version, we have included additional data presented as figures or supplementary figures, to address the questions of the reviewer.

It is difficult to faithfully model the long time course of human atopic march in mice, but we tested to make a longer gap between skin sensitization and airway challenge with a 39-day-protocol (instead 13-day protocol). The results are presented in the attached **Fig3_To_Reviewer2**, showing that even with a longer gap, e.c. or d.c. HDM-sensitized mice still developed allergic asthmatic inflammation upon intranasal challenge, and moreover, asthmatic phenotypes were stronger in d.c. HDM-sensitized mice, compared to e.c. HDM-sensitized mice.

Major Comments:

1. In Figure. 1F to 1J, 2A-C, S2A, 3E-F, 7E, the sample size should be increased. For example, the sample size in the WT d.c. PBS group was only 1 to 2. Furthermore, no significant differences were found in some of the experimental results.

Answer: In **Fig.1, f-i**, and **Fig. 2, a-c**, all the statistic comparisons, i.e. WT/4C13R^{Tg}_ecHDM versus Tslp^{-/-}/4C13R^{Tg}_ecHDM; WT/4C13R^{Tg}_dcHDM versus Tslp^{-/-}/4C13R^{Tg}_dcHDM; WT/4C13R^{Tg}_ecHDM versus WT/4C13R^{Tg}_dcHDM, contain data with n \geq 3 or 4. The data with n=2 were only from the control group WT/4C13R^{Tg}_dcPBS control (without allergen HDM). The reason was that during the experiment one wildtype control mouse accidentally died. As this piece of data served as PBS control, and was not used to calculate statistical significance

with any of other group, we decided to keep the WT/4C13R^{Tg}_dcPBS data as it is. We did not replace WT/4C13R^{Tg}_dcPBS data with data from another experiment, as the current Fig. 1 and 2 represented one experiment containing all the different groups in the same manipulation. Indeed, it is most appropriate to present the data from the one experiment with the age/sex-matched littermates, with the same experimental procedures for treatment and analyses.

Nevertheless, to show the reviewer about WT/4C13R^{Tg}_dcPBS data, we attach here one Figure (**Fig.2_To_Reviewer**) to show one independent experiment which included WT_4C13R^{Tg}_NS; WT/4C13R^{Tg}_ecPBS; WT/4C13R^{Tg}_ecHDM; WT/4C13R^{Tg}_dcPBS; and WT/4C13R^{Tg}_dcHDM. The relationship of these 5 groups including WT/4C13R^{Tg}_dc PBS is similarly observed as **Fig. 1** and **Fig. 2** in the manuscript.

We would like to note that in the revised **Fig.1j**, **Fig. 3g** and **Fig.6e**, we removed the data from non-sensitized groups (WT_4C13R^{Tg}_NS; WT_4C13R^{Tg}_dcPBS), as in these groups without sensitization, HDM-specific IgG1 and IgE were never detected (as expected). For **Fig. 3g**, we replaced the data with another independent experiment which included more experimental mice.

Concerning relatively small numbers in mouse experiments: the composition of experimental mouse cohort is actually limited by the number of appropriate mice available, that is age and sex matched littermates which are produced from breedings. Despite such limitation for mouse number per group, all the data were reproduced with independent experiments. During the revision, we could not re-perform these previous experiments by further increasing the mouse number, particularly due to difficulties and limitations caused by the pandemic, on mouse breeding for the maintenance of mouse lines and the generation of cohort mice,

The significance was calculated with the data obtained. Though certain results do not show $p < 0.05$, the exact p value is always indicated in the figures for those comparison with tendency of difference (e.g. p value close to 0.05).

2. In Figure 1E, 2D, 3D, 4D, and 6D, the authors should evaluate their results using a (semi-)quantitative analysis.

Answer: We thank the reviewer for the suggestion. Cell counts for skin- or lung-infiltrating eosinophils and basophils from immune-stained paraffin sections, related to Fig. 1e, 2d, 3e, 4d are presented now as **Supplementary Fig.1**, to provide a semi-quantitative analysis.

3. In Figures 3 and 4, the authors claimed that TSLP and IL-1 β act independently based on experiments that showed the IL-1 β -enhanced immune response in TSLP-deficient mice. However, they did not mention the possibility that TSLP affected IL-1 β production. For example, the IL-1 β production in TSLP-deficient mice should be assessed when they are subjected to laser-assisted microporation (LMP).

Answer: We thank the reviewer and the other 2 reviewers for the suggestion to clarify the relationship between TSLP and IL-1 β . We have now added new **Fig. 5h** and **Fig. 5i**, showing that TSLP is not required for d.c. HDM-induced IL-1 β or infiltration of neutrophils and monocytes / macrophages, as similar IL-1 β levels and Gr-1 cells were observed from Tslp^{-/-} and wildtype mice; as well as new **Fig. 5j**, showing that the supplement of TSLP does not increase IL-1 β level on e.c. HDM-sensitized skin.

4. In Figure 5A, the authors showed that GR-1hi neutrophils were induced subcutaneously when d.c. was performed, unlike e.c. They should therefore compare e.c. and d.c. to determine the factors that induce subcutaneous infiltration of neutrophils in vivo.

Answer: We performed RT-qPCR analyses to compare the expression of neutrophil-chemoattractant factors in d.c. HDM-treated skin versus e.c. HDM-treated skin. The results are described in **p.12 (Supplementary Fig. 7)**. Among the chemokines examined, CXCL2, CXCL3, CXCL5, CCL3, S100A7A, S100A8 and S100A9 (but not CXCL1, CCL2 or IL-17C) exhibited a significantly increased expression in the skin from d.c. HDM-treated mice compared with e.c. HDM-treated mice. Interestingly, their expression in the epidermis was also higher in d.c. HDM-treated mice than e.c. HDM-treated mice, suggesting that these chemoattractant factors, which are possibly derived (or at least partially) from the epidermis, are potentially implicated in the recruitment of neutrophils in the skin.

5. For a better understanding of the IL-1 β function, it is better to clarify whether IL-1 β acts locally or systemically and how long its expression remains elevated in the skin or body. It will be useful to investigate the dynamics of IL-1 β levels in the blood over time after d.c. sensitization.

Answer: We added a session in the discussion (**p.18**), with **Supplementary Fig. 10**, showing that IL-1 β appears to act rather locally than systemically, as an increase in IL-1 β protein was detected in the skin, but not in a distant organ like lung or in blood circulation at any time points examined upon d.c. HDM treatment.

Minor Comments:

1. The term "GR-1" in the manuscript should be rendered "Gr-1".

Answer: This is corrected.

2. In Figure 4A, "IL-1 β " seems to be incorrectly described as "IL1 β ".

Answer: It is corrected.

3. In Figure 5A, the black line in the FACS results for "Non-treated" seems to be incorrectly included.

Answer: It is corrected. Thank you.

Reviewer #3 (Remarks to the Author):

Major comments

This is a carefully conducted work on the impact of the mode of cutaneous sensitization on the Th2 response to antigen and the subsequent atopic march that leads to allergic lung inflammation.

The experiments are rigorously conducted and the conclusions are strongly backed up by the data presented.

This manuscript advances the field of allergic diseases in general and atopic dermatitis in particular

Answer: We thank very much the reviewer for the enthusiasm for our discoveries.

The manuscript could benefit by addressing the comments below.

Specific comments

1. It would be desirable but not imperative to examine airway hyperresponsiveness in mice differentially exposed to cutaneous sensitization by the e.c. versus d.c. route

Answer: Thanks for this comment. We followed the suggestion to examine the airway function using invasive ventilated lung resistance (R_L) method with flexivent system. The obtained results are summarized:

- 1) in the new **Fig. 2e** (to compare WT and *Tslp*^{-/-} mice sensitized with e.c. HDM or d.c. HDM, followed by i.n. HDM challenge), showing that in WT mice, airway responsiveness to methacholine was increased in e.c. HDM sensitized mice compared with non-sensitized mice, and further enhanced in d.c. HDM sensitization compared to e.c. HDM sensitization. Moreover, the ablation of TSLP diminished the airway hyperresponsiveness in e.c. HDM-sensitized mice but not in d.c. HDM-sensitized mice.
- 2) in the new **Fig. 7e**, showing that in d.c. HDM-sensitized mice, Gr-1 cell depletion led to an attenuated airway hyperresponsiveness, which was restored by co-administration of IL-1 β during the skin sensitization.

These new figures have thus further strengthened our conclusions.

2. The authors report a decrease in IgG1 antibody levels in e.c. sensitized TSLPKO mice compared to WT controls. However, the difference is not statistically significant.

Answer: Indeed, the data showed that HDM-specific IgG1 levels tended to reduce although not reaching statistical significance ($p=0.17$) (**Fig. 1j**). We precise this in the text: “serum levels of HDM-specific IgE and IgG1 in HDM-treated *Tslp*^{-/-} mice were significantly reduced or tended to reduce, respectively”.

3. The authors report a partial effect of *Tslp* deletion on the response to d.c. sensitization. Could this simply be due to the fact that this mode of sensitization results in a more rigorous response. Also, this “partial effect” is equivalent to the effect on e.c. sensitization in some assays such as the percentage of Tfh cells and IL-4⁺Tfh cells.

Answer: In the revised manuscript, we provided more data showing that in the context of d.c. sensitization, TSLP and IL-1 β are two independent factors promoting allergic sensitization and atopic march (see below the answer to the *point-5*). Thus, TSLP and IL-1 β both contribute a partial effect for this mode of sensitization. It is possible this partial effect from TSLP or IL-1 β may not be equal for each parameter, for example, Tfh percentage and IL-4⁺ Tfh cells appear to be more contributed by TSLP, as the decrease of these parameters seems to be both affected by the TSLP KO for d.c. and e.c. sensitization; however, the partial effect of TSLP is clear for other parameters including skin Th2 responses with eosinophils/basophils, BAL cell and airway hyperresponsiveness.

4. The authors have shown that administration of IL-1b does not induces TSLP expression. It would be important to show whether TSLP could induce IL-1b expression

Answer: We thank the reviewer as well as the other two reviewers for this important point. We have now added the new **Fig. 5j**, showing that the supplement of TSLP does not induce IL-1 β level.

5. The authors have shown that e.c. HDM sensitization and to a much larger extent d.c. HDM sensitization induces the infiltration of Il1b expressing GR-1hi and GR-1int cells in the skin. Is this induction dependent of TSLP? This would help strengthen the conclusion that IL-

1b expressing neutrophils and monocytes/macrophages promotes skin inflammation in a TSLP-independent manner.

Answer: Yes, this is an important question. We have added now the new **Fig. 5h** and **Fig. 5i**, showing that TSLP is not required for the induction of IL-1 β or for the infiltration of neutrophils and monocytes / macrophages upon d.c. HDM sensitization, as they were similarly observed in *Tslp*^{-/-} and WT mice. These results have strengthened the conclusion that IL-1 β and TSLP are two independent factors promoting allergic sensitization and atopic march.

6. Mice d.c sensitized with HDM show stronger airway inflammation compared to mice e.c. sensitized with HDM, evidenced by a greater Th2 and eosinophilic response. There were also more neutrophils in the airways of d.c. sensitized mice compared to e.c. sensitized mice. Given the role of neutrophils is severe asthma is the severe airway inflammation in d.c. sensitized mice associated with AHR and is it corticosteroid-sensitive?

Answer: We thank the reviewer to point to the potential link of these data with corticosteroid-resistant asthma. Indeed, **Fig. 2b** shows that neutrophils are higher in d.c. treated mice; **Fig. 4b** shows that the supplement of IL-1 β during e.c HDM increases neutrophil in BAL from both WT and *Tslp*^{-/-} mice. Thus, d.c. HDM or IL-1 β supplement in e.c. HDM both lead to more neutrophils in the airway.

We have added a more detailed paragraph (**p.19**) to summarize and discuss about these results. As pointed by the reviewer, given the recognized role of airway neutrophils in persistent and severe asthma associated with corticoid-resistance, it will be interesting to further investigate in mice whether allergic asthma developed following d.c. sensitization is more resistant to corticosteroid, and to explore in patients the possible link of skin IL-1 β in AD lesions with the development of corticosteroid-resistant asthma with neutrophilic inflammation.

7. Neutrophils are essential for host defense against microbial pathogens by producing antimicrobial molecules. AD patients are often colonized with *S. aureus*. Blocking of IL1b or depleting GR1+ hi or GR1+int cells could be detrimental as it may impair *S aureus* clearance. The authors need to include this potential limitation the Discussion of the potential usefulness of IL-1 blockade and granulocyte depletion strategies in the treatment of asthma in AD patients.

Answer: Yes, we agree with the reviewer that neutrophils and IL-1 β , as many other immune cells or cytokines, are double-edged sword. We added related discussion: **1)** in the paragraph of the **p.20**, we discuss that as AD skin is commonly colonized with *S. aureus*, one may expect that neutrophils and their derived IL-1 β act as double-sided sword, on the one hand contributing to *S. aureus* clearance, and on the other hand, as shown in this study, contributing to the promotion and exacerbation of skin allergic sensitization, which provides one plausible explanation for the association of *S. aureus* in AD with the development of atopic march. **2)** in the last paragraph (**p. 21**), we discuss that given their multiple functions for example IL-1 β in host defence against microbial pathogens, careful design of targeting strategies including administration routes and evaluation of benefits will be certainly required.

We thank again the Reviewers again for their comments and suggestions for improving this manuscript. We hope that this report is now acceptable for publication in the *Nat. Commun.*

REVIEWERS' COMMENTS

Reviewer #1 (Remarks to the Author):

My major concerns have been well addressed, I have no further comments.

Reviewer #2 (Remarks to the Author):

The authors made significant changes to their manuscript by adding the results of several critical experiments. They addressed the majority of the concerns raised by this reviewer. Consequently, the manuscript has been significantly improved.

Reviewer #3 (Remarks to the Author):

The authors have satisfactorily and diligently addressed all my comments